# Understanding Generalization and Forgetting in In-Context Continual Learning

Guangyu Li [1] [*]   Meng Ding [1] [2] [*]   Lijie Hu [1]

## Abstract

In-context learning (ICL) derives its power from enabling Large Language Models to adapt to new tasks via prompt-based reasoning alone, entirely bypassing the need for parameter updates. Existing theories primarily study ICL in single-task settings, while real-world prompts often contain sequences of heterogeneous tasks, leaving a gap in understanding whether Large Language Models implicitly perform continual learning during inference. To bridge this gap, we propose the first theoretical framework for in-context continual learning, modeling how a pretrained Transformer processes multiple sequential tasks within a single prompt through shared attention mechanisms. Focusing on linear and masked linear self-attention, we derive error expressions for model predictions under sequential task prompts and analyze their generalization and forgetting behavior. Our results reveal that standard attention mechanisms inevitably induce inter-task interference by uniformly or causally aggregating historical contexts, leading to systematic bias. We further provide a bias–variance–interference decomposition of prediction error, characterizing when historical in-context information yields positive transfer or provable negative transfer. This analysis exposes fundamental limits of attention-based continual inference and offers theoretical explanations for order sensitivity and performance degradation in long prompts.

## 1. Introduction

Large language models (LLMs) exhibit remarkable generalization abilities across a wide range of tasks, among which *in-context learning* (ICL) has emerged as one of their most prominent features (Brown et al., 2020). Through ICL, pretrained LLMs adapt to a new task purely at inference time using a small set of demonstrations embedded within the prompt, without any parameter updates. This parameter-free adaptation makes LLMs versatile and efficient solvers, tackling diverse problems (Garg et al., 2022; Li et al., 2023; Hu et al., 2024; Jiao et al., 2026; Tang et al., 2026).

The inference-time nature of ICL suggests a tantalizing possibility: beyond one-shot adaptation, could LLMs accumulate, reuse, and update task-relevant information over long contexts, effectively behaving as a *continuous learner*? In practice, real-world prompts are often lengthy and heterogeneous, naturally containing a sequence of tasks rather than a single one. When such prompts are processed by a shared attention mechanism, representations from earlier tasks inevitably influence predictions for later tasks (Kang et al., 2025; Momeni et al., 2025). This interaction can induce phenomena reminiscent of continual learning (CL) (Shi et al., 2025; Zheng et al., 2025), including knowledge transfer, interference, and even negative transfer, yet occurring entirely during inference without parameter updates.

Despite its practical relevance, this phenomenon remains poorly understood. Existing theoretical studies of ICL predominantly focus on single-task settings, where all in-context examples are generated by the same underlying function in one turn (Bai et al., 2023; Ahn et al., 2023; Von Oswald et al., 2023; Zhang et al., 2024b), and thus do not capture inter-task dependencies or sequential interference. Moreover, classical continual learning theory primarily analyzes training-time dynamics driven by parameter updates (Lin et al., 2023; Evron et al., 2022; Ding et al., 2024; Li et al., 2025), which fundamentally differs from the parameter-free, attention-based adaptation intrinsic to ICL. As a result, a critical gap remains between these two lines of research. This gap motivates the following central question:

> *How do LLMs perform continual learning implicitly through attention during inference?*

In this work, we take a first step toward answering this question by developing a theoretical framework for in-context continual learning to understand the generalization and forgetting behaviors in sequential tasks. Our analysis reveals that even in the absence of parameter updates, attention-

---

[*]Equal contribution [1]Mohamed bin Zayed University of Artificial Intelligence (MBZUAI) [2]University of Buffalo. Correspondence to: Lijie Hu <lijie.hu@mbzuai.ac.ae>.

*Proceedings of the 43rd International Conference on Machine Learning*, Seoul, South Korea. PMLR 306, 2026. Copyright 2026 by the author(s).

based aggregation induces systematic bias and interference across tasks, leading to provable limits on continual inference. Our main contributions are summarized as follows:

- **First Theoretical Formalization of In-Context Continual Learning:** We formalize *in-context continual learning* as a setting in which a pretrained Transformer processes a sequence of tasks concatenated into a single prompt, performing sequential task inference through shared attention without parameter updates. This formulation explicitly bridges ICL and core concepts in continual learning, including task order, historical accumulation, and interference.

- **Bias-Variance-Interference Decomposition.** We provide an explicit bias-variance-interference decomposition of prediction error for all specific tasks. This characterization quantifies how task similarity, context length, and the order of tasks jointly determine whether historical context yields positive transfer or provable negative transfer. In particular, we show that increasing context length or task history does not universally improve performance and can degrade generalization when tasks are misaligned.

- **Theoretical Grounding for Empirical Phenomena.** Our framework explains key phenomena observed in real models, such as order-dependent forgetting, the plateauing benefit of longer contexts, and the impact of task similarity on interference. These results provide concrete guidance for designing prompts and evaluating LLM behavior in multi-task or continual settings.

Overall, our results suggest that even without parameter updates, LLMs inherently accumulate, interfere, and sometimes forget task-specific information over long contexts, highlighting both the potential and the limits of attention-driven continual inference and providing concrete guidance for prompt design and multi-task evaluation in practice.

## 2. Related Work

**Empirical studies of synthetic ICL tasks.** Brown et al. (2020) highlighted the exceptional capabilities of the Transformer model in in-context learning within their GPT-3 research. This intriguing phenomenon of Transformers has attracted many attentions, leading to various interpretations and hypotheses about its underlying mechanism (Wei et al., 2022; Olsson et al., 2022; Schaeffer et al., 2023; Chan et al., 2022; Raventós et al., 2023; Min et al., 2022). A recent work by Kang et al. (2025) investigates the ability of in-context learning (ICL) to perform continual learning without parameter updates by leveraging task scheduling and prompt rearrangement. They introduce an in-context continual learning (ICCL) paradigm where large language models demonstrate

long-term retention and reduced catastrophic forgetting in multitask sequences. While this line of work reveals that ICL can emulate continual learning phenomena at inference time, it does not explain when and why in-context generalization succeeds or fails, nor how interference between tasks leads to forgetting, which is the focus of our work.

**Theoretical studies of In-context Learning.** Much theoretical research on ICL centers on the perspective that Transformer models learn specific algorithms during pretraining, which they can then flexibly deploy to solve in-context tasks (Xie et al., 2022; Bai et al., 2023; Li et al., 2024; Liu et al., 2025; Chang et al., 2025). Within this line of research, closest to our work are a series of papers that consider ICL of linear regression by simplified Transformers using linear attention modules (Bai et al., 2023; Ahn et al., 2023; Von Oswald et al., 2023; Zhang et al., 2024b; Wu et al., 2024; Duraisamy, 2024). However, most existing theoretical analyses focus on single-task in-context learning, and few studies examine the behavior of ICL in multi-task sequential settings at inference time. In particular, how generalization performance evolves across a sequence of in-context tasks and how interference between tasks induces forgetting remains poorly understood.

**Continual Learning.** Continual learning (CL) is a learning paradigm where a model needs to continuously learn a sequence of tasks (Shi et al., 2025; Zheng et al., 2025). A primary challenge in continual learning is the *Catastrophic Forgetting*, wherein the model forgets previously acquired knowledge when exposed to new data (Liu & Liu, 2022; Serra et al., 2018; Jin et al., 2021; Momeni et al., 2025; Irie et al., 2025). Based on the huge empirical achievement of continual learning, lots of works aim to analyze the forgetting and generalization error theoretically. Lin et al. (2023) provides explicit theoretical results in a general CL setup with an arbitrary number of tasks, which enables us to comprehensively understand what factors affect the forgetting and generalization performance of CL. Subsequently, a growing body of theoretical work has further deepened these insights, offering more refined analyses of forgetting dynamics and generalization behavior under diverse continual learning settings (Evron et al., 2022; Ding et al., 2024; Li et al., 2025; Li & Hiratani, 2025; Irie et al., 2025).

## 3. Preliminaries

**Notation.** We first describe the notation we use in the paper. Let $[n] = \{1, 2, ..., n\}$. For matrix $X$, we use $[X]_{p:q,r:s}$ to denote the submatrix that contains rows $p$ to $q$ and columns $r$ to $s$, and we use $[X]_{:,i}$ and $[X]_{j,:}$ to denote the i-th column and j-th row of X respectively. We use $I_d$ to denote the $d$-dimensional identity matrix and sometimes we also use $I$ when the dimension is clear from the context. Unless otherwise defined, we use lower case letters for scalars and

vectors, and use upper case letters for matrices.

## 3.1. Transformer and Masked Linear Attention

Following the existing ICL analysis, we focus on a decoder-based Transformer architecture, where each attention layer adopts a decoder-based attention mask (Bai et al., 2023; Ahn et al., 2023; Von Oswald et al., 2023; Zhang et al., 2024b; Wu et al., 2024; Duraisamy, 2024; Zhang et al., 2026). Before formalizing the specific models analyzed, we first recall the definition of the standard softmax self-attention mechanism.

**Definition 3.1** (Masked Multi-Head Self-Attention). Let $D$ denote the embedding dimension and let $P \in \mathbb{R}^{D \times N}$ be an input sequence of length $N$. An $L$-head masked self-attention layer is parameterized by $\{(V_l, Q_l, K_l)\}_{l=1}^L$ with $V_l, Q_l, K_l \in \mathbb{R}^{D \times D}$. The output of the attention layer is

$$\text{Attn}(P) = P + \sum_{l=1}^L (V_l P) \, \text{mask}\big(\sigma\big((K_l P)^\top (Q_l P)\big)\big),$$

where $\sigma(\cdot)$ denotes the softmax function and the mask matrix satisfies

$$[\text{mask}(A)]_{i,j} = \begin{cases} \frac{1}{j} A_{i,j}, & i \leq j, \\ 0, & i > j. \end{cases}$$

Following previous theoretical studies on in-context learning (Zhang et al., 2024a; Lu et al., 2025), we consider two simplified versions of the single-layer self-attention, which are more amenable to theoretical analysis and yet are still capable of in-context learning linear models.

**Linear self-attention.** The linear self-attention model is defined as

$$f_{\text{LSA}}(P;\theta) = P + WP\frac{P^\top VP}{\rho},$$

where $W$ and $V$ are learnable matrices parameterized by $\theta$, and $\rho$ is a normalization factor. This formulation removes the softmax nonlinearity and merges the query–key and projection–value matrices, and has been widely used in theoretical studies of transformers and in-context learning (Zhang et al., 2024a; Ahn et al., 2023; Olsson et al., 2022; Li et al., 2023).

While this model supports in-context learning of linear predictors in single-task scenarios, it mixes information across all tokens uniformly and does not respect task boundaries or order in multi-task prompts. As we show in Theorem 4.1 later, this leads to information leakage from future tasks: the model exploits data from subsequent tasks to predict $y_{t,N+1}$ for the current task $t$.

**Masked linear self-attention.** To encode task order structure, we introduce a causal mask:

$$f_{\text{MSA}}(P;\theta) = P + WP \cdot \text{mask}\big(P^\top VP\big).$$

This model ensures that information flows only forward in the sequence, so that earlier tokens may influence later ones but not vice versa. Compared to linear attention, masked linear attention better aligns with real-world scenarios. Currently, the vast majority of attention-based models adopt this structure and our analysis will primarily focus on this model.

## 3.2. Single Task In-Context Learning

In this work, we focus on in-context learning for regression tasks, where the target labels are continuous and depend on the input features. For an in-context learning task indexed by $t$, a trained Transformer is given $\mathcal{I}_t = (\mathcal{D}_t, x_{t,M+1})$, where $\mathcal{D}_t = \{(x_{t,i}, y_{t,i})\}_{i=1}^M$ consists of in-context examples and $x_{t,M+1}$ is a query[1]. We assume $x_{t,i} \in \mathbb{R}^d$ and $y_{t,i} = f_{w_t}(x_{t,i})$, where $f_{w_t}$ is a deterministic function parameterized by $w_t$. For each ICL instance, $w_t$ is independently sampled from a distribution $P_{w_t}$.

**Definition 3.2** (Data Distribution). For task $t$, inputs follow

$$x_{t,i} \sim \mathcal{N}(0, \Lambda), \qquad y_{t,i} = \langle w_t, x_{t,i} \rangle,$$

and we define

$$\mu_t = \mathbb{E}[x_{t,i} y_{t,i}], \qquad \Sigma_t = \text{Var}(x_{t,i} y_{t,i}).$$

one task-specific prompt corresponds to an embedding matrix $P_t$, defined as

$$P_t := \begin{pmatrix} x_{t,1} & \cdots & x_{t,M} & x_{t,query} \\ y_{t,1} & \cdots & y_{t,M} & 0 \end{pmatrix} \in \mathbb{R}^{D \times (M+1)} \quad (1)$$

Given an embedded prompt matrix $P_t$, the Transformer produces an output sequence $\text{TF}(P_t)$, and a readout function $F$ generates the prediction $\hat{y}_{t,q} = F(\text{TF}(P))$. The goal of ICL is to ensure $\hat{y}_{t,q}$ approximates the target $y_{t,q} = f_{w_t}(x_{t,q})$, i.e., the label set to $0$ in $P_t$.

## 3.3. In-Context Continual Learning

Most existing theoretical analyses of ICL focus on a single task presented within a prompt, where all in-context examples are generated by the same underlying function. In this work, we are interested in understanding how ICL behaves when the prompt contains sequential tasks. In such settings, the language model processes a single long sequence using a shared attention, so information from earlier tasks may influence predictions for later tasks.

This phenomenon naturally connects ICL with continual learning, where task interference and forgetting are central

---

[1] In the subsequent sections, to distinguish examples used for in-context learning, we will denote the query and query label as $x_{t,q}$ and $y_{t,q}$ respectively.

concerns. Traditional continuous learning often suffers from *Catastrophic Forgetting* due to parameter updates during training on different task samples.

We now extend the standard in-context learning setting to a sequential multi-task scenario. A sequence of tasks $t \in \{1, \ldots, T\}$ arrives in order, and all tasks are concatenated into a single prompt processed by a shared attention.

Each task $t$ provides in-context examples $\{(x_{t,i}, y_{t,i})\}_{i=1}^M$ and one query $x_{t,q}$, with $x_{t,i}, x_{t,q} \sim \mathcal{N}(0, \Lambda)$, $y_{t,i} = \langle w_t, x_{t,i} \rangle$. The task parameter $w_t$ is fixed within a task but may vary across tasks.

From the in-context learning perspective, the model must infer the current task parameter $w_t$ from in-context examples and predict $y_{t,q}$ for the query token, without any parameter updates. From the continual learning perspective, the entire concatenated prompt

$$\mathbf{P} = \begin{pmatrix} P_1 & P_2 & \cdots & P_T & Final\_Query \end{pmatrix}$$

is processed by a single attention layer, so representations of earlier tasks may influence predictions for later tasks, potentially causing interference or forgetting. It is noteworthy that we use 'Final_Query' to measure the forgetting metric, which we will formally define shortly. We explicitly denote it as $(x_{\tau,q}, Y_{\tau,q})$, where $\tau$ is an index taking values in $[T]$. We vary $\tau$ to measure the model's learning or retention level for a specific $t$ after learning $T$ tasks. Correspondingly, we denote the prediction label as $\hat{Y}_{\tau,q}$ to distinguish it from $\hat{y}_{t,q}$ following each task.

**Performance Evaluation.** In the single-task ICL setting, performance is typically measured by how well the model predicts the query label based on in-context examples. In the sequential multi-task setting, we need to evaluate not only the prediction accuracy for the current task but also the effect of how much 'knowledge' of old tasks has been forgotten after learning the whole task. Formally, we define:

**Definition 3.3** (Generalization). For task $t$, the generalization error measures how accurately the model predicts the query label from in-context examples:

$$G_t = \mathbb{E}[(\hat{y}_{t,q} - y_{t,q})^2]$$

**Definition 3.4** (Forgetting). For task $t$, we will measure the forgetting after $T$ tasks,

$$F_t = \mathbb{E}[(\hat{Y}_{t,q} - \hat{y}_{t,q})^2]$$

where $\hat{Y}_{\tau,q}$ is the corresponding prediction of $x_{\tau,q}$. It quantifies the increase in error on a previously seen task after the model has processed subsequent tasks.

## 4. Main Results

In this section, we present our main theoretical results and their implications. First, we characterize the prediction error for a specific task $t$, which captures the generalization behavior of in-context continual learning. Second, we study the forgetting behavior across tasks by quantifying how well the model retains the knowledge of each task after observing a total of $T$ tasks.

### 4.1. Generalization

**Lemma 4.1** (Prediction of Linear Self-attention under multiple tasks.). *Consider $T$ tasks, each providing $M$ in-context training examples $\{(x_{t,i}, y_{t,i})\}_{i=1}^M$, and a query input $x_{t,q}$ for task $t$. Under Linear Self-Attention, the prediction is in general, for any task $1 \le t \le T$,*

$$\hat{y}_{t,q} = x_{t,q}^\top \Gamma^{-1} \left( \frac{1}{M} \sum_{s=1}^T \sum_{i=1}^M x_{s,i} y_{s,i} \right).$$

*Remark* 4.2. The proof could be found in Appendix C. Lemma 4.1 reveals a key limitation of standard linear self-attention in the multi-task in-context setting. Since all in-context examples are aggregated uniformly, the resulting prediction does not respect task boundaries. For each task, the prediction error is independent of the task order. While such a formulation is sufficient to analyze single-task in-context learning for linear regression (Li et al., 2023; Ahn et al., 2023; Zhang et al., 2024a), it fails to capture phenomena that arise in multi-task prompts. Furthermore, our experiments in Section 5.1 demonstrate that task order can have a significant impact on generalization error, which cannot be explained by linear self-attention.

This observation motivates the need for a more expressive attention mechanism that can selectively control cross-task information. In the following sections, we introduce masked linear self-attention to address this limitation and enable a principled analysis of generalization and task interference.

**Theorem 4.3** (Prediction Error for Task $t$). *Consider $T$ regression tasks. For task $t$, the model observes $\{(x_{t,i}, y_{t,i})\}_{i=1}^M$ with masked linear self-attention, for $\Gamma = \Lambda + \frac{1}{N}\Lambda + \frac{1}{N}tr(\Lambda)I_d$, the model prediction can be represented as:*

$$\hat{y}_{t,q} = x_{t,q}^\top \Gamma^{-1} \left( \beta_t \sum_{s=1}^t S_s \right),$$

*where $S_s = \sum_{i=1}^M x_{s,i} y_{s,i}$, $\beta_t = \frac{1}{t(M+1)}$. Then the prediction error satisfies*

$$\mathbb{E}[(\hat{y}_{t,q} - y_{t,q})^2] = \min_{w \in \mathbb{R}^d} \mathbb{E}(w^\top x_{t,q} - y_{t,q})^2$$

$$+ \frac{M}{t^2(M+1)^2} \sum_{s=1}^t \mathrm{tr}(\Sigma_s \, \Gamma^{-2} \Lambda)$$

$$+ \sum_{i=1}^d \frac{1}{\lambda_i} \left( \frac{\lambda_i(\alpha s_i - m_i) - m_i \frac{\lambda_i + \mathrm{tr}(\Lambda)}{N}}{\lambda_i + \frac{\lambda_i + \mathrm{tr}(\Lambda)}{N}} \right)^2$$

*Table 1.* Summary of In-Context Continual Learning Dynamics

| Factor | Effect on Generalization (Task $t$) | Effect on Forgetting |
|---|---|---|
| Context Length $M$ | Reduces variance from finite in-context samples; larger $M$ improves intra-task estimation | Variance-induced interference decays as $O(1/M)$; mean-induced interference remains constant for misaligned tasks |
| Task Similarity | High similarity between historical tasks and task $t$ reduces bias; low similarity increases bias term $\|\frac{1}{t}\sum_{s=1}^{t} w_s - w_t\|_2^2$ | Future tasks similar to $t$ increase interference via $\text{tr}(\mu_i \mu_j^\top \Gamma^{-2}\Lambda)$; orthogonal tasks contribute minimally |
| Training Samples $N$ | Leveraging historical info (reduces bias) vs. resisting incorrect transfer (reduces variance/overfitting) | larger $N$ shifts forgetting to other factors; smaller $N$ lets training and sample covariance affect it |
| Task Ordering | N/A in single-task; in multi-task, affects aggregation of what historical tasks | Coefficients $\alpha_i(t)$ explicitly depend on task order; |

*where* $\alpha = \frac{M}{t(M+1)}, s_i = v_i^\top \sum_{s=1}^{t} \mu_s, m_i = v_i^\top \mu_t$ *and* $\Lambda = \sum_{i=1}^{d} \lambda_i v_i v_i^\top$.

**Prediction Error of Task $t$.** Theorem 4.3 provides an explicit characterization of the generalization performance for a specific task $t$, measured by the expected squared prediction error on a fresh query input $x_{t,q}$. We can separate the error into three parts. The first term represents the model's optimal linear irreducible error. The second term stems from the variance introduced by finite samples during inference, which decreases as $M$ increases. The third term is the bias, measuring the deviation between different tasks. When $t = 1$, the bias term reduces to the single-task setting, recovering existing theoretical results on in-context learning for linear regression.

In contrast to linear self-attention, this prediction yields a task-dependent weight $\beta_t$, which enables order-sensitive generalization behavior and lays the foundation for our subsequent analysis of forgetting. Based on Theorem 4.3, we will provide insights on the following aspects.

**(1) *Context Length*. Long context improves generalization only beyond a task-alignment threshold.** When the length of prompts seen during training $N$ is large, $\Gamma^{-1} \approx \Lambda^{-1}$, considering the simplest case that $\Lambda = I_d$, the second error term will be

$$\frac{M}{t^2(M+1)^2}\sum_{s=1}^{t}\text{tr}(\Sigma_s\,\Gamma^{-2}\Lambda) = \frac{(d+1)M}{t^2(M+1)^2}\sum_{s=1}^{t}\|w_s\|_2^2.$$

If we take expectation on $w_i \sim N(0, I_d)$, it is further simplified to $\frac{d(d+1)M}{t(M+1)^2}$. This analysis highlights the role of context length in shaping in-context learning performance, consistent with recent theoretical and empirical studies on ICL (Zhang et al., 2024a; Kang et al., 2025; Lu et al., 2025).

In the single-task setting, a larger $M$ enhances ICL capability and reduces error. In multi-task scenarios, increasing $M$

theoretically does reduce error. However, as we will see in the remaining analysis of task similarity, when tasks are dissimilar, blindly increasing the context length for individual tasks can lead to negative effects. As a result, the benefit of longer context emerges only beyond a certain threshold, whereas below this threshold, additional context can mislead the model and degrade generalization.

**(2) *Task Similarity*. Context helps only when historical tasks are aligned with the target task.** Similarly, when both the test prompt length $M$ and the prompt length observed during training $N$ are large, we have $\Gamma^{-1} \approx \Lambda^{-1}$. Under this regime, the bias term in Theorem 4.3 simplifies to

$$\sum_{i=1}^{d}\frac{1}{\lambda_i}\Big(\frac{\lambda_i(\alpha s_i - m_i) - m_i\frac{\lambda_i + \text{tr}(\Lambda)}{N}}{\lambda_i + \frac{\lambda_i + \text{tr}(\Lambda)}{N}}\Big)^2 = \Big\|\frac{1}{t}\sum_{s=1}^{t}w_s - w_t\Big\|_2^2.$$

When tasks are similar, i.e., $\mu_s$ is close to $\mu_t$, the $\alpha s_i$ is close to $m_i$, rendering the bias term small. In this regime, the overall prediction error is dominated by the variance term, and increasing the number of in-context examples $M$ yields clear performance gains.

In contrast, when tasks differ significantly and historical information is misaligned with the target task, the bias term dominates the error. In this case, simply increasing the context length $M$ does not necessarily improve generalization; instead, it can introduce additional bias.

**(3) *Training Samples*. More training data does not always imply better in-context generalization.** The $\frac{1}{N}$ term in $\Gamma$ acts as an implicit regularization mechanism. By adjusting $N$, the model trades off between historical information to reduce bias and suppressing incorrect transfer caused by task discrepancies, which manifests as increased variance and overfitting to cross-task noise.

When $N$ is sufficiently large ($N \to \infty$), we have $\Gamma \approx \Lambda$, and the bias term is primarily governed by the discrepancy $\alpha s_i - m_i$. In contrast, as $N \to 0^+$, the bias term converges

to $\sum_i m_i^2/\lambda_i$, which depends solely on the target task component and corresponds to ignoring historical information altogether. This behavior indicates that, under our setting, increasing $N$ is not monotonically beneficial.

Indeed, for each coordinate $i$, one can derive an analytical stationary point at which the bias vanishes,

$$N_i^* = \frac{m_i\big(\lambda_i + \mathrm{tr}(\Lambda)\big)}{\lambda_i(\alpha s_i - m_i)}.$$

These stationary points depend on multiple factors, including the input distribution and the alignment between tasks. However, such quantities are typically unknown in advance during training, making it difficult to determine an optimal number of training samples from theoretical considerations alone.

## 4.2. Forgetting

In the following subsection, we provide the interference expression for forgetting and analyze how interference from past and future tasks leads to forgetting in in-context continual learning.

**Theorem 4.4** (Interference Between Task-$t$ Query and Final Query). *After all $T$ tasks, the final prediction is*

$$\hat{Y}_{t,q} = x_{t,q}^\top \Gamma^{-1}\Big(\frac{1}{T(M+1)+1}\sum_{s=1}^{T} S_s\Big).$$

*The difference from the task-t prediction can be written as*

$$\hat{Y}_{t,q} - \hat{y}_{t,q} = x_{t,q}^\top \Gamma^{-1}\Big(\sum_{i=1}^{t} c_t S_i + \sum_{i=t+1}^{T} d S_i\Big),$$

*where*

$$c_t = -\frac{(T-t)M + (T+1-t)}{t(M+1)(TM+T+1)}, \qquad d = \frac{1}{TM+T+1}.$$

*Define*

$$\alpha_i(t) = \begin{cases} c_t, & i \le t, \\ d, & i > t. \end{cases}$$

*We obtain the interference error*

$$\mathbb{E}[(\hat{Y}_{t,q} - \hat{y}_{t,q})^2] = \sum_{i=1}^{T} \alpha_i^2(t)\,\mathrm{tr}\big[(M\Sigma_i + M^2\mu_i\mu_i^\top)\Gamma^{-2}\Lambda\big]$$

$$+ M^2 \sum_{\substack{i,j=1 \\ i\neq j}}^{T} \alpha_i(t)\alpha_j(t)\,\mathrm{tr}(\mu_i\mu_j^\top\Gamma^{-2}\Lambda).$$

*Remark* 4.5. Theorem 4.4 shows that forgetting arises from the **reweighting of task contributions** rather than loss of

information about task $t$ itself. Past tasks ($i \le t$) receive negative coefficients $c_t$, while future tasks ($i > t$) receive positive coefficients $d$. This demonstrates that representations of task $t$ are actively offset by other tasks, reflecting an intrinsic structural property of in-context continual learning.

Furthermore, the above expression reveals two qualitatively distinct sources of interference:

- *intra-task terms*, captured by $\mathrm{tr}(\Sigma_i\Gamma^{-2}\Lambda)$ and $\mathrm{tr}(\mu_i\mu_i^\top\Gamma^{-2}\Lambda)$, which reflect estimation noise within each task;

- *inter-task mean interaction terms*, given by $\mathrm{tr}(\mu_i\mu_j^\top\Gamma^{-2}\Lambda)$ for $i \neq j$, which quantify interference induced by misalignment across different tasks.

Based on Theorem 4.3, we will provide insights on the following aspects.

**(1) *Context Length*. Increasing context length reduces variance-induced interference, but forgetting persists due to irreducible mean misalignment across tasks.** Although the variance-related terms scale linearly with the context length $M$, they are multiplied by coefficients $\alpha_i^2(t) = O(1/M^2)$, resulting in an overall contribution that decays as $O(1/M)$. This implies that increasing context length effectively suppresses estimation noise within each task.

In contrast, all mean-related terms scale as $M^2$ but are weighted by $\alpha_i(t)\alpha_j(t) = O(1/M^2)$. Consequently, their contributions remain bounded and converge to constants as $M$ grows. These limiting values depend on the relative alignment between task means and the task ordering.

This separation highlights a fundamental distinction: while longer contexts improve intra-task estimation accuracy, they cannot eliminate interference arising from task mean misalignment. As a result, forgetting persists asymptotically even in the long-context regime.

**(2) *Task Similarity*. Forgetting is amplified by learning future tasks that are statistically similar to the current one.** Theorem 4.4 reveals that the severity of forgetting is governed by the alignment among task means $\{\mu_i\}$. When task $t$ is similar to subsequent tasks, the interaction term $\mathrm{tr}(\mu_i\mu_j^\top\Gamma^{-2}\Lambda)$ becomes large, leading to strong interference. In contrast, if future tasks are statistically orthogonal to task $t$, their contribution to the interference error is negligible.

This observation indicates that forgetting in in-context continual learning is inherently task-dependent: learning future tasks aligned with past ones amplifies forgetting, whereas learning statistically irrelevant tasks can largely preserve prior performance.

**(3)** *Task Ordering.* **Forgetting is inherently order-dependent, even when the set of tasks remains unchanged.** Since the mixed interaction terms[2] are scaled by $c_t/(TM + T + 1), c_t^2$, their contribution increases with both the number of tasks and the context length. This implies that forgetting is influenced not only by task similarity, but also by the ordering of tasks.

In particular, sequences in which later tasks are misaligned with earlier ones induce stronger inter-task interference. Theorem 4.4 therefore provides a precise mathematical explanation for the order-dependent forgetting phenomena commonly observed in in-context continual learning.

**Corollary 4.6.** *Let $A_i \triangleq \mathrm{tr}\big[(M\Sigma_i + M^2\mu_i\mu_i^\top)\Gamma^{-2}\Lambda\big]$ and $B_{ij} \triangleq M^2\,\mathrm{tr}(\mu_i\mu_j^\top\Gamma^{-2}\Lambda)$. Recall Theorem 4.4 for each $t$, we have $\alpha_i(t)$. For convenience define*

$$\bar{\alpha}_i \triangleq \frac{1}{T}\sum_{t=1}^{T}\alpha_i(t)^2, \qquad \bar{\alpha}_{ij} \triangleq \frac{1}{T}\sum_{t=1}^{T}\alpha_i(t)\alpha_j(t), \quad i \neq j.$$

*Then the average interference error takes*

$$\frac{1}{T}\sum_{t=1}^{T}\mathbb{E}\big[(\hat{Y}_{t,q} - \hat{y}_{t,q})^2\big] = \sum_{i=1}^{T}\bar{\alpha}_i\,A_i + \sum_{\substack{i,j=1 \\ i\neq j}}^{T}\bar{\alpha}_{ij}\,B_{ij}.$$

*Moreover, $S_i$ and $S_{ij}$ admit explicit forms. For each $1 \leq i \leq T$,*

$$\bar{\alpha}_i = \frac{1}{T}\Big((i-1)d^2 + \sum_{t=i}^{T}c_t^2\Big).$$

*For $i < j$,*

$$\bar{\alpha}_{ij} = \frac{1}{T}\Big((i-1)d^2 + \sum_{t=i}^{j-1}c_t d + \sum_{t=j}^{T}c_t^2\Big),$$

*and for $i > j$, symmetry gives $\bar{\alpha}_{ij} = \bar{\alpha}_{ji}$.*

While the previous results characterize interference at a fixed task-query pair, continual learning typically evaluates forgetting by averaging performance across tasks. The corollary provides the average forgetting in ICL. The averaged error aligns with the standard definition of forgetting in continual learning (Evron et al., 2022; Lin et al., 2023), where performance degradation is evaluated by aggregating over tasks rather than conditioning on a specific task.

The different coefficients $\bar{\alpha}_i, \bar{\alpha}_{ij}$ reveal that task ordering influences forgetting not only at specific query positions but also in an average sense across all tasks. Unlike $\alpha_i(t)$, which depends on a particular query task $t$, these averaged

coefficients are independent of the query position and thus quantify the typical contribution of each task to forgetting across the entire learning process, indicating that interference is a global property of in-context continual learning.

## 4.3. Summary of Generalization and Forgetting Insights

The preceding analysis of masked linear self-attention in in-context continual learning allows us to summarize key factors affecting generalization and forgetting.

**Proposition 4.7** (Summary of In-Context Continual Learning Dynamics)**.** *Let $M$ be the number of in-context examples per task, $N$ the prompt length seen during training, and $\{\mu_i\}$ the task means. We summarize the effects of various factors on generalization and forgetting in Table 1.*

***Remark.*** *This proposition consolidates the insights from Theorems 4.3 and 4.4, as well as Corollary B.3. Generalization errors decompose into irreducible, variance, and bias terms, which are affected by context length, task similarity, and training prompt length. Forgetting emerges structurally from reweighting of task contributions, and persists asymptotically when task means are misaligned, regardless of context length.*

## 5. Experiments

In this section, we empirically validate our theoretical findings using more sophisticated transformer models. We study how context length, task similarity and task ordering affect in-context continual learning, and assess the extent to which our predictions about generalization and forgetting align with empirical performance.

**Experimental Setup.** We utilize a standard decoder-style GPT-2 architecture consisting of 12 layers, 8 attention heads, and an embedding dimension of 256. Furthermore, to validate the scalability of our conclusions across different model sizes, we conduct additional verifications on a tiny GPT-2 configuration (3 layers, 2 heads, embedding dimension of 64). We evaluate these transformers on linear regression tasks with mean-zero Gaussian inputs and fixed identity covariance, following the training procedure of Garg et al. (2022) where the model learns in-context learning from diverse task instances.

Each task is defined by a task-specific weight vector $w \in \mathbb{R}^d$, and task identity is implicit, determined by example ordering in the prompt sequence. During inference, all model parameters remain fixed, and adaptation occurs solely through the attention mechanism. We generate multiple sequences of tasks and report mean squared error (MSE) on query points, including per-task prediction error, forgetting error and average interference error, averaged over multiple batches with standard deviations. Further experimental details, including model architectures, convergence losses, and comprehen-

---

[2] For details on which specific task interactions correspond to different coefficients, please refer to Corollary B.3

sive tabular results across various scales, are provided in Appendix G.

## 5.1. Effect of Context Length $M$

According to Theorem 4.3, the variance term should shrink for fixed task position $t$. Increasing $M$ also reduces estimation noise in the sample covariance, improving the effective estimator used by the attention mechanism. However, the bias term also reveals that beyond a certain regime, increasing context length can amplify systematic inter-task interference, potentially worsening generalization when task means are misaligned.

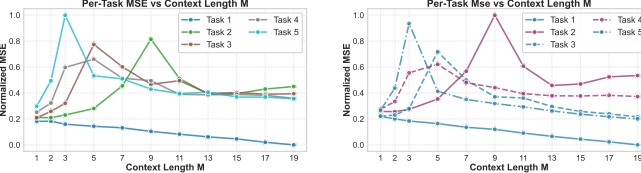

*Figure 1.* Per-Task MSE vs Context Length M.

**Results.** *The effect of context length $M$ on prediction error is task-dependent, reflecting a trade-off between variance reduction and inter-task interference.* In the single-task setting (Task 1), increasing the number of in-context samples $M$ consistently reduces the prediction error, reflecting the standard variance reduction under finite-sample estimation. In the multi-task setting, however, the effect of $M$ is no longer monotonic (Figure 2). Prediction error results from a trade-off between variance reduction from additional in-context examples and bias induced by task dissimilarity and inter-task interference. When $M$ is small, interference dominates, and increasing $M$ can even increase error; as $M$ grows, variance reduction becomes dominant, yielding improved performance.

Consistent with our theoretical predictions, the variance term decreases with $M$ approximately as $\mathcal{O}(M/(M+1)^2)$ for larger $M$, whereas the interference term persists, depending on the alignment of task-specific parameters. This explains why the benefit of increasing $M$ eventually plateaus.

To systematically validate our theoretical bias-variance-interference decomposition (Theorem 4.3), we performed a quantitative comparison between the theoretically predicted MSE and the empirical MSE actually observed in the GPT-2 model. As detailed in Appendix G, the structural inflection points are highly consistent despite architectural differences. For subsequent tasks, the empirical error exhibits a significant peak at intermediate context lengths (e.g., at $M = 3$) before recovering. This directly validates our claim: when historical tasks are misaligned, expanding context initially injects systematic bias that overwhelms variance reduction.

Figure 2 further illustrates the role of task similarity. Among the five tasks, $\{1, 3, 5\}$ form one similarity cluster and $\{2, 4\}$ form another. Tasks 3, 4, and 5 benefit from their similarity to preceding tasks, resulting in lower error, a phenomenon reminiscent of data-replay methods in continual learning (Rolnick et al., 2019; Ding et al., 2026).

## 5.2. Task similarity

While the previous experiment varied the amount of historical context, we now isolate the role of task similarity by holding the context length fixed. Task similarity directly controls the magnitude of the bias term: when tasks are well aligned, previous context reduces error through variance reduction, whereas misaligned tasks introduce systematic interference. To systematically control this, we parameterize the similarity by the angle $\theta$ between the task-specific weight vectors.

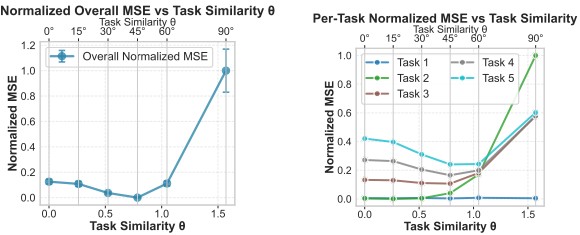

*Figure 2.* Per-Task MSE vs Task Similarity.

**Results.** Our results clearly validate the theoretical bias-variance trade-off. When $\theta$ is small, corresponding to high task similarity, historical tasks provide positive transfer and reduce prediction error. As $\theta$ increases, the bias term grows and systematic interference dominates, leading to negative transfer where incorporating dissimilar historical tasks increases the MSE. These results demonstrate that task similarity is a decisive factor in determining whether historical context is beneficial or harmful, consistent with the bias characterization in Theorem 4.3.

Furthermore, this controlled setting provides a crucial theoretical bridge to understanding realistic LLM behaviors. While our synthetic construction allows for a smooth interpolation of $\theta$, heterogeneous natural language tasks are typically mapped to nearly orthogonal representations in the embedding space after large-scale pretraining. This inherently places real-world multi-task prompting in the large $\theta$ regime of our framework, mathematically explaining why systematic interference and negative transfer naturally dominate when dealing with diverse NLP tasks.

## 5.3. Task Order Sensitivity

Theorem 4.4 reveals that forgetting in in-context learning arises from attention reweighting rather than information

loss. Because adaptation occurs entirely during a single forward pass without parameter updates, the entire sequence of tasks is perfectly preserved in the context window. Thus, forgetting intrinsically arises from how the attention mechanism aggregates this history. While the mask provides order, it induces systematic bias by aggregating historical tasks with fixed, decaying weights.

**Results.** Our experiments demonstrate clear order-dependent forgetting: tasks that appear earlier in the sequence experience greater forgetting after processing subsequent tasks, consistent with the theoretical prediction that attention weights decay with distance. We observe that the forgetting pattern depends on both the task position and the specific task ordering, validating the interference mechanism described in Theorem 4.4. The results show that forgetting arises intrinsically from attention-based aggregation, even without any parameter updates, confirming that in-context continual learning exhibits genuine forgetting behavior.

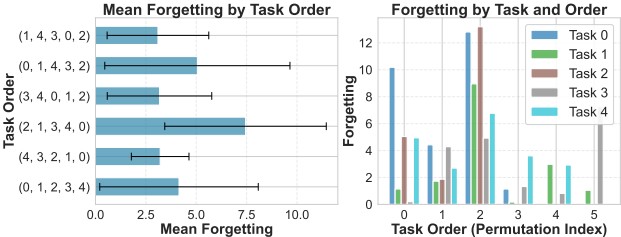

*Figure 3.* Task Order vs. Forgetting

### 5.4. Generalization to Non-linear Tasks

To demonstrate that our theoretical insights are not limited to simplified linear settings, we extend our experiments to more complex function classes, including sparse linear regression and two-layer ReLU neural networks ($f(x) = \sum_{i=1}^{r} \alpha_i \sigma(w_i^\top x)$).

As detailed in the comprehensive tabular results in Appendix G, increasing the context length $M$ consistently reduces prediction error for the first task. However, for subsequent tasks, increasing $M$ initially improves performance but eventually causes the error to rise sharply. For instance, in sparse linear regression, the normalized MSE for Task 5 jumps catastrophically to 0.9905 at $M = 19$. This explicitly confirms our theoretical finding that aggregating extended historical contexts from heterogeneous tasks inevitably induces systematic bias and provable negative transfer. It holds robustly even in highly non-linear environments.

### 5.5. Validation on Real-world LLMs

To validate our results in realistic multi-task prompting scenarios, we designed an ICCL evaluation pipeline using a modern instruction-tuned large language model, Qwen2.5-

1.5B-Instruct (Team et al., 2024). We selected two heterogeneous text classification tasks: SST-2 (binary sentiment classification (Socher et al., 2013), Task A) and AG News (four-class topic classification (Zhang et al., 2015), Task B). We constructed prompts by concatenating $M$ in-context examples from Task A, followed by $M$ examples from Task B, requiring the model to generate predictions in a single forward pass.

*Table 2.* ICCL Evaluation on Qwen2.5-1.5B-Instruct.

| $M$ | Task B (Negative Transfer) | | | Task A (Forgetting) | | |
| --- | --- | --- | --- | --- | --- | --- |
| | Baseline | ICCL | $\Delta$ | Baseline | Final | $\Delta$ |
| 1 | 0.736 | 0.580 | -0.156 | 0.934 | 0.472 | -0.462 |
| 3 | 0.702 | 0.630 | -0.072 | 0.938 | 0.468 | -0.470 |
| 5 | 0.716 | 0.670 | -0.046 | 0.948 | 0.496 | -0.452 |
| 19 | 0.668 | 0.684 | +0.016 | 0.922 | 0.536 | -0.386 |

Table 2 summarizes the phenomena of negative transfer and forgetting. When $M$ is small, the historical context from Task A heavily biases the attention mechanism, causing severe negative transfer that drops Task B's accuracy by over 15% compared to its baseline. As $M$ increases, the variance reduction effect of the relevant context emerges, allowing the model to gradually isolate task interference and recover to baseline performance levels by $M = 19$.

Furthermore, evaluating the model's retention of Task A at the very end of the concatenated prompt demonstrates catastrophic degradation. Even though the LLM's weights were entirely frozen, accuracy on Task A dropped from 0.934 to 0.472 at $M = 1$ after introducing Task B. Although increasing $M$ mitigates some variance-induced interference, a severe baseline level of forgetting persists asymptotically due to the irreducible mean misalignment between the two distinct NLP tasks, exactly as predicted by Theorem 4.4.

Comprehensive tabular results, detailed experimental setups, and further ablation studies across different architectures are provided in Appendix G.

## 6. Conclusion

In this work, we introduce the first theoretical framework for in-context continual learning, analyzing how shared attention processes task sequences. By quantifying prediction error in masked linear self-attention, we identify conditions for positive versus negative transfer. We show that attention aggregation inherently causes interference—specifically order sensitivity, bias, and forgetting—even without parameter updates. Validated by experiments, these findings reveal the fundamental limits of inference-time adaptation and necessitate principled strategies for update-free continual learning.

## Impact Statement

This paper presents work whose goal is to advance the field of Machine Learning. There are many potential societal consequences of our work, none of which we feel must be specifically highlighted here.

## Acknowledgements

This work is supported in part by the MBZUAI Research Fund BF0100. We also thank the anonymous reviewers for their valuable feedback and constructive suggestions.

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

# A. Pre-training Procedure

In this work, we consider the task of in-context learning linear predictors. Following the assumptions of Zhang et al. (2024a), we employed the same training design, including embedding matrices and initialization to obtain results regarding model parameter convergence under gradient flow.

We will assume training prompts are sampled as follows. Let $\Lambda$ be a positive definite covariance matrix. Each training prompt, indexed by $\tau \in \mathbb{N}$, takes the form of $P_\tau = (x_{\tau,1}, h_\tau(x_{\tau_1}), \ldots, x_{\tau,N}, h_\tau(x_{\tau,N}), x_{\tau,q})$, where task weights $w_\tau \overset{iid}{\sim} \mathcal{N}(0, I_d)$, inputs $x_{\tau,i}, x_{\tau,q} \overset{iid}{\sim} \mathcal{N}(0, \Lambda)$, and labels $h_\tau(x) = w_\tau^\top x$.

Each prompt corresponds to an embedding matrix $E_\tau$, formed using the transformation (1):

$$E_\tau := \begin{pmatrix} x_{\tau,1} & x_{\tau,2} & \cdots & x_{\tau,N} & x_{\tau,q} \\ w_\tau^\top x_{\tau,1} & w_\tau^\top x_{\tau,2} & \cdots & w_\tau^\top x_{\tau,N} & 0 \end{pmatrix} \in \mathcal{R}^{(d+1)\times(N+1)}.$$

The empirical risk over $B$ independent prompts is defined as

$$\widehat{L}(\theta) = \frac{1}{2B} \sum_{\tau=1}^{B} \left( \widehat{y}_{\tau,q} - w_\tau^\top x_{\tau,q} \right)^2. \tag{2}$$

We consider the behavior of gradient flow-trained networks over the population loss induced by the limit of infinite training tasks $B \to \infty$:

$$L(\theta) = \lim_{B\to\infty} \widehat{L}(\theta) = \frac{1}{2}\mathbb{E}_{w_\tau, x_{\tau,1}, \cdots, x_{\tau,N}, x_{\tau,q}} \left[ (\widehat{y}_{\tau,q} - w_\tau^\top x_{\tau,q})^2 \right] \tag{3}$$

Above, the expectation is taken w.r.t. the covariates $\{x_{\tau,i}\}_{i=1}^{N} \cup \{x_q\}$ in the prompt and the weight vector $w_\tau$, i.e. over $x_{\tau,i}, x_q \overset{iid}{\sim} \mathcal{N}(0, \Lambda)$ and $w_\tau \sim \mathcal{N}(0, I_d)$. Gradient flow captures the behavior of gradient descent with an infinitesimal step size and has dynamics given by the following differential equation:

$$\frac{\mathrm{d}}{\mathrm{d}t}\theta = -\nabla L(\theta). \tag{4}$$

We will consider gradient flow with an initialization that satisfies the following and cite their theorem here. More details please refer to Zhang et al. (2024a).

**Assumption A.1** (Initialization). Let $\sigma > 0$ be a parameter, and let $\Theta \in \mathbb{R}^{d\times d}$ be any matrix satisfying $\|\Theta\Theta^\top\|_F = 1$ and $\Theta\Lambda \neq 0_{d\times d}$. We assume

$$W(0) = \sigma \begin{pmatrix} 0_{d\times d} & 0_d \\ 0_d^\top & 1 \end{pmatrix}, \quad V(0) = \sigma \begin{pmatrix} \Theta\Theta^\top & 0_d \\ 0_d^\top & 0 \end{pmatrix} \tag{5}$$

**Theorem A.2** (Convergence and limits). *Consider the gradient flow of the linear self-attention network defined in Section 3.1. Suppose the initialization satisfies Assumption A.1 with initialization scale $\sigma > 0$ satisfying $\sigma^2 \|\Gamma\|_{op} \sqrt{d} < 2$ where we have defined*

$$\Gamma := (1 + \frac{1}{N})\Gamma + \frac{1}{N}tr(\Lambda)I_d \in \mathbb{R}^{d\times d}$$

*Then the gradient flow converges to a global minimum of the population loss. Moreover, $W$ and $V$ converge to $W^*$ and $V^*$ respectively, where*

$$W^* = a^{\frac{1}{4}} \begin{pmatrix} 0_{d\times d} & 0_d \\ 0_d^\top & 1 \end{pmatrix}, \quad V^* = a^{-\frac{1}{4}} \begin{pmatrix} \Gamma^{-1} & 0_d \\ 0_d^\top & 0 \end{pmatrix} \tag{6}$$

*where $a = tr(\Gamma^{-2})$.*

*Remark* A.3 (Training Effect). In our theoretical analysis, the effect of training is primarily reflected in the final learned parameter matrix, which we characterize in Theorem A.2. As discussed in Section 4.1, the dominant factor influencing this process is the training samples.

Taking generalization as an example, in Section 4.1 we provide a detailed analysis of how the number of samples affects model performance. In general, larger sample sizes do not always lead to better performance. In our experimental observations, however, with the number of training steps set to 500k, the final loss tends to stabilize regardless of the number

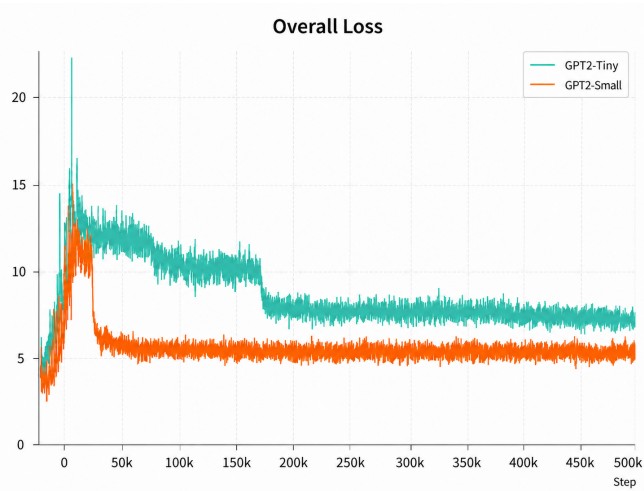

*Figure 4.* Training Loss Curve

of training samples, as long as it exceeds a certain threshold. To further support our theoretical findings, we present the training loss curve, which shows that under the curriculum learning we used in the initial training step, the loss initially decreases and then increases as the number of samples grows.

We also emphasize that the obtained parameter is derived under gradient flow dynamics with appropriate initialization, yielding a closed-form analytical result. In practical training scenarios, however, the learning process is affected by additional factors, such as the learning rate, model size and the number of attention heads, etc. We leave it to future work.

## B. Useful Lemma and Corollary

**Lemma B.1.** *Define $S_s = \sum_{i=1}^{M} x_{s,i} y_{s,i}$ and $\mu_t = \mathbb{E}[x_{t,i} y_{t,i}], \Sigma_t = \mathrm{Var}(x_{t,i} y_{t,i})$, then we have*

$$\mathbb{E}[S_i S_j^\top] = \begin{cases} M\Sigma_i + M^2 \mu_i \mu_i^\top, & i = j, \\ M^2 \mu_i \mu_j^\top, & i \neq j, \end{cases}$$

**Lemma B.2.** *For $x \sim \mathcal{N}(\mu, \Lambda), \Lambda \in \mathbb{R}^{d \times d}$ and a matrix $A \in \mathbb{R}^{d \times d}$, we have the first four moment as follows:*

$$\mathbb{E}(x) = \mu,$$
$$\mathbb{E}(xx^\top) = \Lambda,$$
$$\mathbb{E}(\|w\|^2) = \|\mu\|^2 + \mathrm{Tr}(\Lambda),$$

**Corollary B.3** (Forgetting Decomposition). *Separating the Theorem 4.4 from tasks $1 \leq i \leq t$ and $t < i \leq T$ yields*

$$\mathbb{E}[(\hat{Y}_{t,q} - \hat{y}_{t,q})^2]$$
$$= Mc_t^2 \sum_{i=1}^{t} \mathrm{tr}(\Sigma_i \Gamma^{-2} \Lambda) + \frac{M}{(TM + T + 1)^2} \sum_{i=t+1}^{T} \mathrm{tr}(\Sigma_i \Gamma^{-2} \Lambda)$$
$$+ M^2 c_t^2 \sum_{i=1}^{t} h_{i,i} + \frac{M^2}{(TM + T + 1)^2} \sum_{i=t+1}^{T} h_{i,i}$$
$$+ M^2 c_t^2 \sum_{\substack{i,j=1 \\ i \neq j}}^{t} h_{i,j} + \frac{M^2}{(TM + T + 1)^2} \sum_{\substack{i,j=t+1 \\ i \neq j}}^{T} h_{i,j}$$
$$\frac{M^2 c_t}{TM + T + 1} \sum_{1 \leq i < t < j \leq T} h_{i,j} + \frac{M^2 c_t}{TM + T + 1} \sum_{1 \leq j < t < i \leq T} h_{i,j}$$

*where $h_{i,j} = \mathrm{tr}(\mu_i \mu_j^\top \Gamma^{-2} \Lambda)$ induces an inner product between task means.*

**Theorem B.4** (Prediction Error for Task $t$ after T Tasks). *We take the same notation in Theorem 4.3, then the prediction error for task $t$ satisfies*

$$\mathbb{E}[(\hat{Y}_{t,q} - y_{t,q})^2] = \min_{w \in \mathbb{R}^d} \mathbb{E}(w^\top x_{t,q} - y_{t,q})^2$$

$$+ \frac{M}{(TM + T + 1)^2} \sum_{s=1}^{T} \operatorname{tr}(\Sigma_s \, \Gamma^{-2} \Lambda)$$

$$+ \sum_{i=1}^{d} \frac{1}{\lambda_i} \Big( \frac{\lambda_i(\beta s_i - m_i) - m_i \frac{\lambda_i + \operatorname{tr}(\Lambda)}{N}}{\lambda_i + \frac{\lambda_i + \operatorname{tr}(\Lambda)}{N}} \Big)^2$$

*where $\beta = \frac{M}{TM+T+1}, \Lambda = \sum_{i=1}^{d} \lambda_i u_i u_i^\top, s_i = u_i^\top \sum_{s=1}^{T} \mu_s, m_i = u_i^\top \mu_t$.*

**Corollary B.5** (Average Prediction Error after T Tasks).

$$\frac{1}{T} \sum_{t=1}^{T} \mathbb{E}\big[(\hat{Y}_{t,q} - y_{t,q})^2\big] = \frac{1}{T} \sum_{t=1}^{T} \min_{w \in \mathbb{R}^d} \mathbb{E}(w^\top x_{t,q} - y_{t,q})^2$$

$$+ \frac{M}{(TM + T + 1)^2} \sum_{s=1}^{T} \operatorname{tr}(\Sigma_s \, \Gamma^{-2} \Lambda)$$

$$+ \sum_{i=1}^{d} \frac{1}{\lambda_i} \Big[ (\bar{m}_i - \frac{\lambda_i \beta s_i}{d_i})^2 + \frac{1}{T} \sum_{t=1}^{T} (m_{i,t} - \bar{m}_i)^2 \Big]$$

*where $d_i = \lambda_i + \frac{\lambda_i + \operatorname{tr}(\Lambda)}{N}, m_{i,t} = u_i^\top \mu_t, \bar{m}_i = \frac{1}{T} \sum_{t=1}^{T} m_{i,t}$.*

## C. Proof of Lemma 4.1

*Proof.* Recall the definition of Linear Self-Attention,

$$f_{\text{LSA}}(P; \theta) = P + WP \frac{P^\top V P}{\rho},$$

so, we have

$$\hat{y}_{t,q} = [f_{\text{LSA}}(P; \theta)]_{(d+1),(t(M+1))}.$$

Since the prediction takes only the right-bottom entry of the token matrix output by the LSA layer, actually only part of $W$ and $V$ affect the prediction. To see how, let us denote

$$W = \begin{pmatrix} W_{11} & w_{12} \\ w_{21}^\top & w_{22} \end{pmatrix} \in \mathbb{R}^{(d+1) \times (d+1)}, \qquad V = \begin{pmatrix} V_{11} & v_{12} \\ v_{21}^\top & v_{22} \end{pmatrix} \in \mathbb{R}^{(d+1) \times (d+1)}$$

where $W_{11} \in \mathbb{R}^{d \times d}, w_{12}, w_{21} \in \mathbb{R}^d, w_{22} \in \mathbb{R}$ and $V_{11} \in \mathbb{R}^{d \times d}, v_{12}, v_{21} \in \mathbb{R}^d, v_{22} \in \mathbb{R}$. Then the prediction query is

$$\hat{y}_{t,q} = \begin{pmatrix} w_{21}^\top & w_{22} \end{pmatrix} \cdot (\frac{PP^\top}{M}) \begin{pmatrix} V_{11} \\ v_{21}^\top \end{pmatrix} x_{t,query},$$

Since only the last row of $W$ and the first d columns of $V$ affect the prediction, which means we can simply take all other entries zero in the following sections, corresponding to Assumption A.1.

Here, the inner product of the embedding matrix $P$ is

$$PP^\top = \begin{pmatrix} P_1 & P_2 & \cdots & P_T & q_\tau \end{pmatrix} \begin{pmatrix} P_1^\top \\ P_2^\top \\ \ldots \\ P_T^\top \\ q_\tau^\top \end{pmatrix} = P_1 P_1^\top + \cdots + q_\tau q_\tau^\top$$

where

$$P_t P_t^\top = \begin{pmatrix} \frac{1}{M}\sum_{i=1}^{M} x_{t,i}x_{t,i}^\top + \frac{1}{M}x_{t,q}x_{t,q}^\top & \frac{1}{M}\sum_{i=1}^{M} x_{t,i}y_{t,i} \\ \frac{1}{M}\sum_{i=1}^{M} x_{t,i}^\top y_{t,i} & \frac{1}{M}\sum_{i=1}^{M} y_{t,i}^2 \end{pmatrix} \in \mathbb{R}^{(d+1)\times(d+1)}.$$

Based on Theorem A.2, the prediction $\hat{y}_{t,q}$ is

$$\hat{y}_{t,q} = \begin{pmatrix} 0_d^\top & 1 \end{pmatrix} \left[ \begin{pmatrix} \frac{1}{M}\sum_{i=1}^{M} x_{1,i}x_{1,i}^\top + \frac{1}{M}x_{1,q}x_{1,q}^\top & \frac{1}{M}\sum_{i=1}^{M} x_{1,i}y_{1,i} \\ \frac{1}{M}\sum_{i=1}^{M} x_{1,i}^\top y_{1,i} & \frac{1}{M}\sum_{i=1}^{M} y_{1,i}^2 \end{pmatrix} + \cdots \right] \begin{pmatrix} \Gamma^{-1} & 0_d \\ 0_d^\top & 0 \end{pmatrix} \begin{pmatrix} x_{t,q} \\ 0 \end{pmatrix}$$

$$= x_{t,q}^\top \Gamma^{-1} \left( \frac{1}{M}\sum_{i=1}^{M} x_{1,i}y_{1,i} + \cdots \right)$$

$$= x_{t,q}^\top \Gamma^{-1} \left( \frac{1}{M}\sum_{s=1}^{T}\sum_{i=1}^{M} x_{s,i}y_{s,i} \right).$$

$\square$

# D. Proof of Theorem 4.3 and Corollary B.5

*Proof.* We divide the proof into two parts. In the first part, we will prove the prediction of Mask Linear Self-attention and in the second part we will provide the prediction error for a specific $t$.

Recall the definition of Mask Linear Self-Attention,

$$f_{\mathrm{MSA}}(P;\theta) = P + WP \cdot \mathrm{mask}(P^\top V P).$$

To process the mask matrix, we write it in vector form,

$$[f_{\mathrm{MSA}}]_i = p_i + \frac{1}{i}\sum_{j=1}^{i} p_i^\top V p_j \cdot W p_j.$$

where $p_i$ is the i-th column of the embedding matrix of $P$. So, we have

$$\begin{pmatrix} * \\ \hat{y}_{t,q} \end{pmatrix} = \begin{pmatrix} x_{t,q} \\ 0 \end{pmatrix} + \frac{1}{t(M+1)}\sum_{j=1}^{t(M+1)} \begin{pmatrix} x_{t,j}^\top & y_{t,j} \end{pmatrix} \begin{pmatrix} \Gamma^{-1} & 0_d \\ 0_d^\top & 0 \end{pmatrix} \begin{pmatrix} x_{t,q} \\ 0 \end{pmatrix} \begin{pmatrix} \mathbf{0}_{d\times d} & 0_d \\ 0_d^\top & 1 \end{pmatrix} \begin{pmatrix} x_{t,j} \\ y_{t,j} \end{pmatrix}$$

$$\hat{y}_{t,q} = x_{t,q}^\top \Gamma^{-1} \left( \frac{1}{t(M+1)}\sum_{s=1}^{t}\sum_{j=1}^{M} x_{s,j}y_{s,j} \right)$$

That is the prediction of MSA module, different from LSA module below. In the following, we characterize the excess risk of the prediction of a trained LSA layer with respect to the risk of the best linear predictor, on a new task which is possibly non-linear.

Unless otherwise specified, we denote $\mathbb{E}$ as the expectation over $(x_{t,i}, y_{t,i}), (x_{t,q}, y_{t,q}) \sim \mathcal{D}$. Sine when $(x, y) \sim \mathcal{D}$, we assume that $\mathbb{E}(x), \mathbb{E}(y), \mathbb{E}(xx^\top), \mathbb{E}(yxx^\top y)$ exist. We denote

$$a = \mathrm{argmin}_{w\in\mathbb{R}^d} \mathbb{E}(w_t x_{t,q}^\top - y_{t,q})^2, \mu_t = \mathbb{E}[x_{t,i}y_{t,i}], \Sigma_t = \mathrm{Var}(x_{t,i}y_{t,i}).$$

as the weight of the best linear estimation. Actually, if we denote the function inside the minimum above as $R(w)$, we can write it as

$$R(w_t) = w_t^\top \Lambda w_t - 2\mathbb{E}(y_{t,q} \cdot x_{t,q}^\top)w_t + \mathbb{E}(y_{t,q}^2).$$

Since the Hessian matrix $\frac{\partial^2}{\partial w_t \partial w_t^\top}R(w_t)$ is $\Lambda$, which is positive definite, we know that this function is strictly convex and hence, the global minimum can be reached at the unique first-order stationary point. That is

$$a = \Lambda^{-1}\mathbb{E}(x_{t,q}y_{t,q}).$$

We also define a similar vector for ease of computation:

$$b = \Gamma^{-1} \frac{M}{t(M+1)} \sum_{s=1}^{t} \mathbb{E}(x_{s,q} y_{s,q}) = \Gamma^{-1} \frac{M}{t(M+1)} \sum_{s=1}^{t} \mu_s.$$

Therefore, we can decompose the error as

$$\begin{aligned}
\mathbb{E}[(\hat{y}_{t,q} - y_{t,q})^2] &= \mathbb{E}(\hat{y}_{t,q} - b^\top x_{t,q})^2 + \mathbb{E}(b^\top x_{t,q} - a^\top x_{t,q}) \\
&\quad + \mathbb{E}(a^\top x_{t,q} - y_{t,q})^2 + 2\mathbb{E}(\hat{y}_{t,q} - b^\top x_{t,q})(a^\top x_{t,q} - y_{t,q}) \\
&\quad + 2\mathbb{E}(\hat{y}_{t,q} - b^\top x_{t,q})(b^\top x_{t,q} - a^\top x_t, q) + 2\mathbb{E}(y_{t,q} - a^\top x_{t,q})(b^\top x_{t,q} - a^\top x_{t,q})
\end{aligned}$$

The first term is the first term on the Theorem 4.3. So it suffices to calculate others.

From the tower property of conditional expectation, we have

$$\begin{aligned}
\mathbb{E}(\hat{y}_{t,q} - b^\top x_{t,q})(b^\top x_{t,q} - a^\top x_{t,q}) &= \mathbb{E}\left[\mathbb{E}\left((\hat{y}_{t,q} - b^\top x_{t,q})(b^\top x_{t,q} - a^\top x_{t,q})|x_{t,q}\right)\right] \\
&= \mathbb{E}\left[\mathbb{E}\left((\hat{y}_{t,q} - b^\top x_{t,q})|x_{t,q}\right)(b^\top x_{t,q} - a^\top x_{t,q})\right] = 0,
\end{aligned}$$

since

$$\mathbb{E}\left((\hat{y}_{t,q} - b^\top x_{t,q})|x_{t,q}\right) = \left(\mathbb{E}\frac{1}{t(M+1)}\Gamma^{-1}\left(\sum_{s=1}^{t}\sum_{j=1}^{M} x_{s,j} y_{s,j}\right) - b\right)^\top x_{t,q} = 0.$$

Similarly, we have

$$\begin{aligned}
\mathbb{E}(\hat{y}_{t,q} - b^\top x_{t,q})(a^\top x_{t,q} - y_{t,q}) &= \mathbb{E}\left[\mathbb{E}\left((\hat{y}_{t,q} - b^\top x_{t,q})(a^\top x_{t,q} - y_{t,q})|x_{t,q}, y_{t,q}\right)\right] \\
&= \mathbb{E}\left[\mathbb{E}\left((\hat{y}_{t,q} - b^\top x_{t,q})|x_{t,q}, y_{t,q}\right)(a^\top x_{t,q} - y_{t,q})\right] = 0.
\end{aligned}$$

Beyond them, we also have

$$\begin{aligned}
\mathbb{E}(y_{t,q} - a^\top x_{t,q})(b^\top x_{t,q} - a^\top x_{t,q}) &= \mathbb{E}\operatorname{tr}\left[(y_{t,q} - a^\top x_{t,q})(b-a)^\top x_{t,q}\right] \\
&= \operatorname{tr}[(b-a)a^\top \Lambda] - \operatorname{tr}[(b-a)\mathbb{E}(x_{t,q}y_{t,q})] = 0.
\end{aligned}$$

where the last comes from the definition of $a$. Therefore, all cross terms vanish and it suffices to consider the remaining.

Under the property of trace and the fact that $\Gamma$ and $\Lambda$ commute, We have

$$\begin{aligned}
&\mathbb{E}\left(\frac{1}{t(M+1)}\sum_{s=1}^{t} S_s - \frac{M}{t(M+1)}\sum_{s=1}^{t}\mu_s\right)\Gamma^{-1} x_{t,q} x_{t,q}^\top \Gamma^{-1}\left(\frac{1}{t(M+1)}\sum_{s=1}^{t} S_s - \frac{M}{t(M+1)}\sum_{s=1}^{t}\mu_s\right) \\
&= \mathbb{E}\operatorname{tr}\left(\frac{1}{t(M+1)}\sum_{s=1}^{t} S_s - \frac{M}{t(M+1)}\sum_{s=1}^{t}\mu_s\right)\left(\frac{1}{t(M+1)}\sum_{s=1}^{t} S_s - \frac{M}{t(M+1)}\sum_{s=1}^{t}\mu_s\right)^\top \Gamma^{-2}\Lambda \\
&= \frac{1}{t^2(M+1)^2}\sum_{i,j=1}^{M}\mathbb{E}\operatorname{tr}\left\{(\sum_{s=1}^{t} x_{s,i}y_{s,i} - \sum_{s=1}^{t}\mu_s)(\sum_{s=1}^{t} x_{s,j}y_{s,j} - \sum_{s=1}^{t}\mu_s)^\top \Gamma^{-2}\Lambda\right\} \\
&= \frac{M}{t^2(M+1)^2}\mathbb{E}\operatorname{tr}\left\{(\sum_{s=1}^{t} x_{s,1}y_{s,1} - \sum_{s=1}^{t}\mu_s)(\sum_{s=1}^{t} x_{s,1}y_{s,1} - \sum_{s=1}^{t}\mu_s)^\top \Gamma^{-2}\Lambda\right\} \\
&= \frac{M}{t^2(M+1)^2}\sum_{s=1}^{t}\operatorname{tr}(\Sigma_s \Gamma^{-2}\Lambda).
\end{aligned}$$

Notice that all cross terms vanish due to the independence of $x_i$, and the last line comes from the definition of $\Sigma$.

For the last term, we have

$$\mathbb{E}(b-a)^\top x_{t,q} x_{t,q}^\top (b-a) = \mathrm{tr}\left[(b-a)^\top \Lambda (b-a)\right].$$

We consider the quadratic form

$$\mathcal{Q} \triangleq \left(\Gamma^{-1}\alpha S - \Lambda^{-1}\mu_t\right)^\top \Lambda\left(\Gamma^{-1}\alpha S - \Lambda^{-1}\mu_t\right), \text{ where } \alpha = \frac{M}{t(M+1)}, \ S = \sum_{s=1}^{t}\mu_s,$$

where

$$\Gamma = \left(1+\frac{1}{N}\right)\Lambda + \frac{\mathrm{tr}(\Lambda)}{N}I_d.$$

Since $\Gamma$ is a polynomial in $\Lambda$, the two matrices commute and are simultaneously diagonalizable. Let

$$\Lambda = U\,\mathrm{diag}(\lambda_1,\ldots,\lambda_d)U^\top,$$

and define the coordinates

$$s_i = u_i^\top S, \qquad m_i = u_i^\top \mu_t, \qquad \tau = \mathrm{tr}(\Lambda) = \sum_{j=1}^{d}\lambda_j.$$

Then

$$\Gamma = U\,\mathrm{diag}(c\lambda_i + d)U^\top, \quad c = 1+\frac{1}{N}, \quad d = \frac{\tau}{N}.$$

Using orthogonality of $U$, the quadratic form decomposes as

$$\mathcal{Q} = \sum_{i=1}^{d}\left[\frac{\alpha^2\lambda_i}{(c\lambda_i+d)^2}s_i^2 - \frac{2\alpha}{c\lambda_i+d}s_i m_i + \frac{1}{\lambda_i}m_i^2\right].$$

Combining the terms over a common denominator yields

$$\mathcal{Q} = \sum_{i=1}^{d}\frac{\alpha^2\lambda_i^2 s_i^2 - 2\alpha\lambda_i(c\lambda_i+d)s_i m_i + (c\lambda_i+d)^2 m_i^2}{\lambda_i(c\lambda_i+d)^2}.$$

Note that

$$c\lambda_i + d = \frac{(N+1)\lambda_i + \tau}{N} = \frac{\lambda_i N + (\lambda_i + \tau)}{N}.$$

Multiplying numerator and denominator by $N^2$ gives

$$\mathcal{Q} = \sum_{i=1}^{d}\frac{N^2\alpha^2\lambda_i^2 s_i^2 - 2\alpha N\lambda_i\left(\lambda_i N + \lambda_i + \tau\right)s_i m_i + \left(\lambda_i N + \lambda_i + \tau\right)^2 m_i^2}{\lambda_i\left(\lambda_i N + \lambda_i + \tau\right)^2}.$$

The numerator is a perfect square:

$$N^2\alpha^2\lambda_i^2 s_i^2 - 2\alpha N\lambda_i\left(\lambda_i N + \lambda_i + \tau\right)s_i m_i + \left(\lambda_i N + \lambda_i + \tau\right)^2 m_i^2 = \left(\lambda_i(\alpha s_i - m_i)N - (\lambda_i + \tau)m_i\right)^2.$$

Hence,

$$\mathcal{Q} = \sum_{i=1}^{d}\frac{\left(\lambda_i(\alpha s_i - m_i)N - (\lambda_i + \tau)m_i\right)^2}{\lambda_i\left(\lambda_i N + (\lambda_i + \tau)\right)^2}.$$

Equivalently,

$$\mathcal{Q} = \sum_{i=1}^{d}\frac{1}{\lambda_i}\left(\frac{\lambda_i(\alpha s_i - m_i)N - (\lambda_i + \tau)m_i}{\lambda_i N + (\lambda_i + \tau)}\right)^2,$$

which is an exact expression exhibiting the dependence on $N$. So, overall, the prediction error satisfies

$$\mathbb{E}[(\hat{y}_{t,q} - y_{t,q})^2] = \min_{w \in \mathbb{R}^d} \mathbb{E}(w^\top x_{t,q} - y_{t,q})^2 + \frac{M}{t^2(M+1)^2} \sum_{s=1}^{t} \operatorname{tr}(\Sigma_s \, \Gamma^{-2} \Lambda) + \sum_{i=1}^{d} \frac{1}{\lambda_i} \left( \frac{\lambda_i(\alpha s_i - m_i) - m_i \frac{\lambda_i + \operatorname{tr}(\Lambda)}{N}}{\lambda_i + \frac{\lambda_i + \operatorname{tr}(\Lambda)}{N}} \right)^2,$$

where $\alpha = \frac{M}{t(M+1)}, s_i = u_i^\top \sum_{s=1}^{t} \mu_s, m_i = u_i^\top \mu_t$ and $\Lambda = \sum_{i=1}^{d} \lambda_i u_i u_i^\top$.

For Corollary B.5, we proceed by diagonalizing $\Lambda$. Let

$$\Lambda = U \operatorname{diag}(\lambda_1, \ldots, \lambda_d) U^\top, \quad s_i = u_i^\top \sum_{s=1}^{T} \mu_s, \quad m_{i,t} = u_i^\top \mu_t, \quad \tau = \operatorname{tr}(\Lambda).$$

Then $\Gamma = \left(1 + \frac{1}{N}\right) \Lambda + \frac{\tau}{N} I_d$, so in each eigen-direction $u_i$ we have

$$d_i = \lambda_i + \frac{\lambda_i + \tau}{N}.$$

Hence for each $t$ and $i$, the contribution in the $i$-th eigen-direction is

$$Q_{i,t} = \frac{1}{\lambda_i} \left( \frac{\lambda_i \beta s_i}{d_i} - m_{i,t} \right)^2.$$

$$\frac{1}{T} \sum_{t=1}^{T} Q_{i,t} = \frac{1}{T} \sum_{t=1}^{T} \left( \frac{\lambda_i \beta s_i}{d_i} - m_{i,t} \right)^2.$$

Let $\bar{m}_i = \frac{1}{T} \sum_{t=1}^{T} m_{i,t}$. Then

$$\frac{1}{T} \sum_{t=1}^{T} \left( \frac{\lambda_i \beta s_i}{d_i} - m_{i,t} \right)^2 = \frac{1}{T} \sum_{t=1}^{T} \left( \frac{\lambda_i \beta s_i}{d_i} - \bar{m}_i + \bar{m}_i - m_{i,t} \right)^2$$

$$= \frac{1}{T} \sum_{t=1}^{T} \left( \left( \frac{\lambda_i \beta s_i}{d_i} - \bar{m}_i \right)^2 + 2 \left( \frac{\lambda_i \beta s_i}{d_i} - \bar{m}_i \right)(\bar{m}_i - m_{i,t}) + (\bar{m}_i - m_{i,t})^2 \right)$$

$$= \left( \frac{\lambda_i \beta s_i}{d_i} - \bar{m}_i \right)^2 + \frac{1}{T} \sum_{t=1}^{T} (m_{i,t} - \bar{m}_i)^2.$$

The cross term vanishes because $\sum_{t=1}^{T} (m_{i,t} - \bar{m}_i) = 0$.

Finally, summing over $i = 1, \ldots, d$, we obtain

$$\frac{1}{T} \sum_{t=1}^{T} R_t = \sum_{i=1}^{d} \frac{1}{\lambda_i} \left[ \left( \bar{m}_i - \frac{\lambda_i \beta s_i}{d_i} \right)^2 + \frac{1}{T} \sum_{t=1}^{T} (m_{i,t} - \bar{m}_i)^2 \right].$$

Hence, the full average prediction error becomes

$$\frac{1}{T} \sum_{t=1}^{T} \mathbb{E}\left[ (\hat{Y}_{t,q} - y_{t,q})^2 \right] = \frac{1}{T} \sum_{t=1}^{T} \min_{w \in \mathbb{R}^d} \mathbb{E}(w^\top x_{t,q} - y_{t,q})^2$$

$$+ \frac{M}{(TM + T + 1)^2} \sum_{s=1}^{T} \operatorname{tr}(\Sigma_s \Gamma^{-2} \Lambda) + \sum_{i=1}^{d} \frac{1}{\lambda_i} \left[ \left( \bar{m}_i - \frac{\lambda_i \beta s_i}{d_i} \right)^2 + \frac{1}{T} \sum_{t=1}^{T} (m_{i,t} - \bar{m}_i)^2 \right].$$

$\square$

# E. Proof of Theorem 4.4 and Corollary B.3

*Proof.* Similar to $\hat{y}_{t,q}$, we have

$$\begin{pmatrix} * \\ \hat{Y}_{t,q} \end{pmatrix} = \begin{pmatrix} x_{t,q} \\ 0 \end{pmatrix} + \frac{1}{T(M+1)+1} \sum_{j=1}^{T(M+1)} \begin{pmatrix} x_{t,j}^\top & y_{t,j} \end{pmatrix} \begin{pmatrix} \Gamma^{-1} & 0_d \\ 0_d^\top & 0 \end{pmatrix} \begin{pmatrix} x_{t,q} \\ 0 \end{pmatrix} \begin{pmatrix} \mathbf{0}_{d\times d} & 0_d \\ 0_d^\top & 1 \end{pmatrix} \begin{pmatrix} x_{t,j} \\ y_{t,j} \end{pmatrix}$$

$$\hat{Y}_{t,q} = x_{t,q}^\top \Gamma^{-1} \Big( \frac{1}{T(M+1)+1} \sum_{s=1}^{T} \sum_{j=1}^{M} x_{s,j} y_{s,j} \Big).$$

Comparing the two estimations, we have

$$\hat{Y}_{t,q} - \hat{y}_{t,q} = x_{t,q}^\top \Gamma^{-1} \Big( \sum_{i=1}^{t} c_i S_i + d \sum_{i=t+1}^{T} S_i \Big),$$

where

$$c_i = -\frac{(T-t)M + T + 1 - t}{t(M+1)(T+M+1)}, \quad d = \frac{1}{T(M+1)+1}.$$

Let

$$B = \sum_{i=1}^{t} c_i S_i + d \sum_{i=t+1}^{T} S_i.$$

Then

$$(\hat{Y}_{t,q} - \hat{y}_{t,q})^2 = (x_{t,q}^\top \Gamma^{-1} B)^2 = \mathrm{tr}\big( B B^\top \Gamma^{-1} x_{t,q} x_{t,q}^\top \Gamma^{-1} \big).$$

Taking expectation and using iterated expectation on $x_{t,q}$,

$$\mathbb{E}\big[ (\hat{Y}_{t,q} - \hat{y}_{t,q})^2 \big] = \mathrm{tr}\big( \mathbb{E}[B B^\top] \Gamma^{-2} \Lambda \big).$$

where $\Gamma$ and $\Lambda$ are commute.

Writing $B = \sum_{i=1}^{T} \alpha_i S_i$ with $\alpha_i = c_i$ for $i \le t$ and $\alpha_i = d$ for $i > t$, we have

$$\mathbb{E}[B B^\top] = \sum_{i=1}^{T} \sum_{j=1}^{T} \alpha_i \alpha_j \, \mathbb{E}[S_i S_j^\top].$$

If $S_i$ are sums of i.i.d. terms with means $\mu_i$ and covariances $\Sigma_i$, and groups are independent. Following the Lemma B, we have

$$\mathbb{E}[S_i S_j^\top] = \begin{cases} M\Sigma_i + M^2 \mu_i \mu_i^\top, & i = j, \\ M^2 \mu_i \mu_j^\top, & i \ne j. \end{cases}$$

Hence

$$\mathbb{E}\big[ (\hat{Y}_{t,q} - \hat{y}_{t,q})^2 \big] = \sum_{i=1}^{T} \alpha_i^2 \Big( M \, \mathrm{tr}(\Sigma_i \Gamma^{-2} \Lambda) + M^2 \, \mathrm{tr}(\mu_i \mu_i^\top \Gamma^{-2} \Lambda) \Big) + \sum_{i \ne j} \alpha_i \alpha_j \, M^2 \, \mathrm{tr}(\mu_i \mu_j^\top \Gamma^{-2} \Lambda).$$

The proof of Corollary B.3 is evident; one needs only unfold the interactions between different tasks according to the task sequence.

$\square$

# F. Proof of Corollary 4.6

*Proof.* Recall that the interference error at task $t$ is given by

$$\mathbb{E}\big[(\hat{Y}_{t,q} - \hat{y}_{t,q})^2\big] = \sum_{i=1}^{T} \alpha_i(t)^2 \operatorname{tr}\big[(M\Sigma_i + M^2 \mu_i \mu_i^\top)\Gamma^{-2}\Lambda\big] + M^2 \sum_{i \neq j} \alpha_i(t)\alpha_j(t) \operatorname{tr}(\mu_i \mu_j^\top \Gamma^{-2}\Lambda),$$

where

$$\alpha_i(t) = \begin{cases} c_t, & i \leq t, \\ d, & i > t, \end{cases} \qquad d = \frac{1}{TM + T + 1}.$$

We consider the time-averaged interference error

$$\frac{1}{T}\sum_{t=1}^{T} \mathbb{E}\big[(\hat{Y}_{t,q} - \hat{y}_{t,q})^2\big].$$

Substituting the above expression and exchanging the order of summation yields

$$\frac{1}{T}\sum_{t=1}^{T} \mathbb{E}\big[(\hat{Y}_{t,q} - \hat{y}_{t,q})^2\big] = \sum_{i=1}^{T} S_i \operatorname{tr}\big[(M\Sigma_i + M^2 \mu_i \mu_i^\top)\Gamma^{-2}\Lambda\big] + M^2 \sum_{i \neq j} S_{ij} \operatorname{tr}(\mu_i \mu_j^\top \Gamma^{-2}\Lambda),$$

where the averaged coefficients are defined as

$$S_i \triangleq \frac{1}{T}\sum_{t=1}^{T} \alpha_i(t)^2, \qquad S_{ij} \triangleq \frac{1}{T}\sum_{t=1}^{T} \alpha_i(t)\alpha_j(t), \quad i \neq j.$$

For a fixed task index $i$, note that $\alpha_i(t) = d$ for $t < i$ and $\alpha_i(t) = c_t$ for $t \geq i$. Hence,

$$\sum_{t=1}^{T} \alpha_i(t)^2 = \sum_{t=1}^{i-1} d^2 + \sum_{t=i}^{T} c_t^2 = (i-1)d^2 + \sum_{t=i}^{T} c_t^2,$$

which gives

$$S_i = \frac{1}{T}\Big((i-1)d^2 + \sum_{t=i}^{T} c_t^2\Big).$$

Without loss of generality, assume $i < j$. The product $\alpha_i(t)\alpha_j(t)$ admits the following partition:

$$\alpha_i(t)\alpha_j(t) = \begin{cases} d^2, & t < i, \\ c_t d, & i \leq t < j, \\ c_t^2, & t \geq j. \end{cases}$$

Therefore,

$$\sum_{t=1}^{T} \alpha_i(t)\alpha_j(t) = \sum_{t=1}^{i-1} d^2 + \sum_{t=i}^{j-1} c_t d + \sum_{t=j}^{T} c_t^2,$$

and hence

$$S_{ij} = \frac{1}{T}\Big((i-1)d^2 + \sum_{t=i}^{j-1} c_t d + \sum_{t=j}^{T} c_t^2\Big).$$

By symmetry, the same expression holds for $i > j$.

Combining the expressions of $S_i$ and $S_{ij}$ completes the derivation of the time-averaged interference error and proves the claim.

$\square$

# G. Additional Experiments Results and Setting Details

**Model Architecture.** We use decoder-style GPT2 transformers trained on linear regression tasks with mean-zero Gaussian inputs. Each task is associated with a fixed identity covariance matrix across prompts. The training procedure follows Garg et al. (2022), where the model learns to perform in-context learning by observing diverse task instances. During inference, the model parameters remain fixed, and adaptation occurs solely through the attention mechanism.

**Data Generation.** Each task corresponds to a linear regression problem: input vectors $x \in \mathbb{R}^d$ are drawn from a standard Gaussian distribution $\mathcal{N}(0, I_d)$, and task-specific weight vectors $w \in \mathbb{R}^d$ define the linear mapping. Task identity is implicit, determined solely by the ordering of examples within the prompt sequence.

**Training Hyperparameters** The synthetic experiments utilized two scales of decoder-style GPT-2 models. The models were trained from scratch using the Adam optimizer. The detailed hyperparameters are provided below:

- **Standard GPT-2**: 12 layers, 8 attention heads, and an embedding dimension of 256 ($\sim 9.5$M parameters). The model was trained with a learning rate of $1 \times 10^{-4}$ and a batch size of 64 for 500k steps.

- **Tiny GPT-2**: 3 layers, 2 attention heads, and an embedding dimension of 64 ($\sim 0.2$M parameters). The model was trained using similar optimization settings. The final converged training loss refers to Figure 4.

**Evaluation.** For each experiment, we generate multiple sequences of tasks and measure the model's prediction error on query points. We report the mean squared error (MSE) averaged over multiple batches, along with standard deviations. The evaluation metrics directly correspond to the theoretical quantities introduced in Sections 3.3 and 4, including per-task prediction error, forgetting error (the difference between intermediate-task predictions and final predictions after processing all tasks), and the average interference error across all tasks.

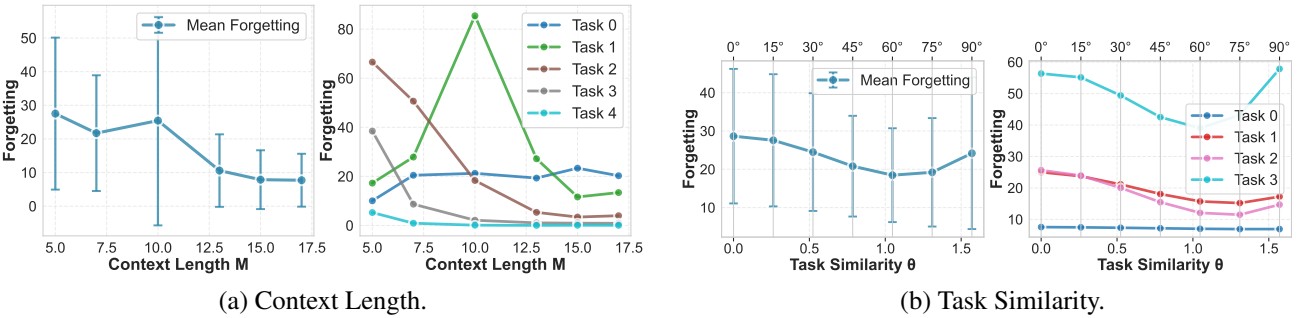

(a) Context Length.        (b) Task Similarity.

*Figure 5.* Different factors influence forgetting.

## G.1. Context Length

**Experimental design.** We fix the number of tasks $T = 5$ and vary the number of in-context examples per task $M \in \{1, 2, 3, 5, \dots, 20\}$. For each value of $M$, we evaluate the prediction error for each task position and compute the overall average MSE.

In addition to presenting in the main text how $M$ affects generalization behavior, we here provide supplementary results on how context length influences forgetting, as illustrated in Figure 5. As discussed in the main text, increasing the context length effectively reduces variance-induced interference, but forgetting remains due to irreducible interference arising from mean misalignment across tasks.

## G.2. Task Similarity

**Experimental design.** We construct task parameters with controlled similarity by generating task weight vectors $w_t$ such that the angle between task means is controlled by a parameter $\theta$. We fix the number of tasks $T = 5$ and the context length $M = 20$, and vary $\theta \in \{0, \pi/12, \pi/6, \pi/4, \pi/3, \pi/2\}$. To ensure fair comparison, we use the same set of fixed task weights across all $\theta$ values, only varying the relationship between tasks.

*Table 3.* Complete Normalized MSE for Non-linear Tasks across varying Context Lengths ($M$).

| Task Type | $M$ | Task 1 | Task 2 | Task 3 | Task 4 | Task 5 |
|---|---|---|---|---|---|---|
| **Sparse Linear** | 1 | 0.1709 | 0.1728 | 0.1979 | 0.2240 | 0.2433 |
| | 2 | 0.1564 | 0.1929 | 0.2555 | 0.3038 | 0.3842 |
| | 3 | 0.1267 | 0.2075 | 0.3220 | 0.3937 | 0.4173 |
| | 5 | 0.0880 | 0.2629 | 0.3933 | 0.3714 | 0.3483 |
| | 7 | 0.0504 | 0.2907 | 0.3625 | 0.3206 | 0.3278 |
| | 9 | 0.0235 | 0.3003 | 0.2993 | 0.3192 | 0.8545 |
| | 11 | 0.0091 | 0.2818 | 0.2974 | 0.7724 | 0.9095 |
| | 13 | 0.0053 | 0.2537 | 0.7137 | 0.8742 | 0.9182 |
| | 15 | 0.0015 | 0.2334 | 0.7347 | 0.8773 | 0.9787 |
| | 19 | 0.0000 | 0.2255 | 0.8571 | 1.0000 | 0.9905 |
| **2-layer ReLU NN** | 1 | 0.5334 | 0.6092 | 0.6264 | 0.6723 | 0.7179 |
| | 2 | 0.4414 | 0.5887 | 0.6964 | 0.8010 | 0.8480 |
| | 3 | 0.3693 | 0.5726 | 0.7137 | 0.8356 | 0.8905 |
| | 5 | 0.2992 | 0.6407 | 0.7726 | 0.9099 | 0.9677 |
| | 7 | 0.2246 | 0.5856 | 0.8496 | 0.8851 | 0.9494 |
| | 9 | 0.1657 | 0.6196 | 0.8154 | 0.9527 | 0.9506 |
| | 11 | 0.0995 | 0.5666 | 0.8178 | 0.8963 | 0.9268 |
| | 13 | 0.0900 | 0.6166 | 0.8007 | 0.8868 | 0.9881 |
| | 15 | 0.0478 | 0.5639 | 0.7575 | 0.9403 | 0.9032 |
| | 19 | 0.0000 | 0.5861 | 0.7313 | 0.8367 | 1.0000 |

In addition to the main text, where we show how task similarity affects generalization behavior, we here provide supplementary results on its impact on forgetting in Figure 5. As discussed, forgetting is amplified when future tasks are statistically similar to the current task.

### G.3. Task Ordering

**Experimental design.** We evaluate prediction error for intermediate tasks both before and after processing subsequent tasks. Specifically, for a sequence of $T$ tasks, we measure the prediction error for task $t$ at two points: (1) immediately after task $t$ is presented (before processing tasks $t + 1, \ldots, T$), and (2) after all $T$ tasks have been processed. We also compare different task orderings while keeping the task set fixed, using permutations of the same task weights to isolate the effect of order from task identity.

### G.4. More Complex Tasks

**Experimental design.** To verify that the systematic bias and inter-task interference observed in our linear framework extend to highly non-linear environments, we evaluated the models on Sparse Linear Regression and two-layer ReLU Neural Networks. The two-layer ReLU network is formally defined as $f(x) = \sum_{i=1}^{r} \alpha_i \sigma(w_i^\top x)$. We maintained the same sequence structure of $T = 5$ tasks and varied the context length $M$. For evaluation, the normalized MSE is computed by applying global min-max scaling across all evaluated context lengths and tasks within the same task family.

**Extended Results.** Table 3 presents the complete set of normalized MSE values across all 5 tasks and 10 different context lengths. Consistent with the theoretical predictions in the main text, increasing $M$ initially provides variance reduction (visible in Task 1). However, for subsequent tasks (Tasks 3, 4, and 5), incorporating misaligned historical context introduces severe systematic bias, causing the error to rise sharply at larger context lengths.

### G.5. Real-world LLMs: Prompting and Evaluation Details

**Experimental setup.** To validate our theory in realistic NLP scenarios, we utilized the HuggingFace 'transformers' library to evaluate the `Qwen2.5-1.5B-Instruct` model on a sequence of heterogeneous text classification tasks. We selected SST-2 (binary sentiment classification) as Task A and AG News (four-class topic classification) as Task B. For few-shot

demonstrations, we sampled examples from the respective training splits, while the evaluation queries were drawn from the SST-2 validation split and the AG News test split. We utilized a standard causal language modeling generation pipeline with `max_new_tokens=32`.

**Prompt Construction.** We concatenated $M$ historical examples from Task A, followed by $M$ historical examples from Task B, and appended the final target query. Task identity was implicitly provided by the demonstration format. The precise prompt template structure is as follows:

---

**Prompt Template for ICCL Evaluation**

```
You are an expert text classifier. Read the examples and classify
the final text accordingly.

--- Task 1: Sentiment Analysis ---
Text: [SST-2 Text]
Label: [Positive / Negative]

... (M examples of Task A) ...

--- Task 2: News Topic Classification ---
Text: [AG News Text]
Label: [World / Sport / Business / Science]

... (M examples of Task B) ...

--- Task [1 or 2]: [Task Name] ---
Text: [Target Query Text]
Label:
```

---

**Evaluation Metric.** Rather than relying on log-probabilities, we extracted the model's prediction directly from the generated free-text completion. We mapped the output to the corresponding class ID through robust sub-string keyword matching (e.g., matching "positive"/"negative" for SST-2, and "world"/"sport"/"business"/"science"/"tech" for AG News). Accuracy was then computed against the ground-truth labels.

