# OpenReview forum: "Understanding Generalization and Forgetting in In-Context Continual Learning"
_ICML.cc/2026/Conference — ICML 2026 regular_

### Official Review · Reviewer_pUNh · 2026-03-11

**Soundness:** 3
**Presentation:** 3
**Significance:** 2
**Originality:** 2
**Overall Recommendation:** 4
**Confidence:** 4

**Summary:**

This paper investigates the intersection of in-context learning (ICL) and continual learning by studying generalization and forgetting in a sequential, multi-task setting. The authors analyze how models behave when tasks are presented sequentially in-context, characterizing intra-task learning dynamics and inter-task interference.

**Compliance With Llm Reviewing Policy:**

Affirmed.

**Final Justification:**

After reading the author reply, the reviewer understands the problem difficulty and raises the rating.

**Key Questions For Authors:**

The authors can review the limitation points below.

**Limitations:**

Limited Novelty: The primary contribution—combining ICL with continual learning in a sequential setting—is conceptually incremental. The paper claims to "consider multiple tasks," but this is the standard definition of continual learning. Simply applying this established paradigm to an ICL context, without introducing a new problem formulation or a novel mechanism to address it, does not constitute a sufficient contribution. The paper reads more as a direct application of existing continual learning ideas to ICL rather than a novel synthesis that yields new insights.

Expected Insights: The main findings regarding intra-task learning and inter-task interference are largely predictable. In any sequential learning system, one would expect to see performance improve within a task (intra-task) and degrade on previous tasks when new ones are introduced (inter-task interference/catastrophic forgetting). The paper does not convincingly demonstrate what new understanding of ICL is gained from this analysis, beyond confirming that ICL models exhibit behaviors already well-documented in other continual learning contexts.

Overly Restrictive Theoretical Framework: The theoretical analysis relies on a highly simplified model: a one-layer self-attention mechanism and linear models. Given that the power of ICL in practice emerges from deep, nonlinear architectures, it is unclear what the derived results imply for realistic scenarios. The paper does not adequately address the technical challenge of extending this analysis to more complex models, nor does it explain how the combination of ICL and continual learning introduces new technical difficulties that are not present when studying each field in isolation. The gap between the theoretical claims and the practical systems they aim to understand is simply too wide.

**Strengths And Weaknesses:**

The paper combines two active areas of research—ICL and continual learning—by framing task sequences within a single context window. It attempts to provide theoretical insights into how generalization and forgetting manifest in this novel setting and quantifies the interactions between tasks presented sequentially.

While the paper addresses an interesting intersection of topics, several significant weaknesses undermine its contribution. The core idea lacks novelty, the insights are largely expected, and the theoretical analysis rests on simplifying assumptions that limit its applicability to realistic ICL scenarios.

---

> ### Author Rebuttal · Authors · 2026-03-31
>
> **Q1. Limited Novelty: The primary contribution—combining ICL with continual learning in a sequential setting—is conceptually incremental...**
>
> A1.Thanks for the comment. However, we respectfully disagree that our contribution is merely a direct application of existing continual learning (CL) ideas. Classical CL theory predominantly analyzes forgetting driven by parameter updates in the training phase(i.e., weight overwriting) [1,2,3]. In contrast, our work provides the first theoretical formalization of how continual learning and forgetting occur implicitly during inference, entirely without parameter updates. This shifts the fundamental mechanism of forgetting from physical information loss to attention-based aggregation.
>
> To address this novel formulation, we derived a bias-variance-interference decomposition (Theorems 4.3 and 4.4) that explicitly quantifies how shared attention processes sequential tasks. This framework provides the first rigorous proof of how inter-task interference and negative transfer manifest purely from in-context attention dynamics,  bridging a critical theoretical gap that neither single-task in-context learning theories nor classical parameter-update continual learning theories can explain.
>
> * [1] Lin et al, "Theory on forgetting and generalization of continual learning." NeurIPS, 2023.
> * [2] Ding et al, "Understanding forgetting in continual learning with linear regression." ICML, 2024.
> * [3] Li et al, "Towards understanding catastrophic forgetting in two-layer convolutional neural networks." ICML, 2025.
>
> ---
>
> **Q2. Expected Insights: The main findings regarding intra-task learning and inter-task interference are largely predictable...**
>
> A2. Thanks for the comment. However, We respectfully disagree that these insights are merely predictable confirmations. While macroscopic phenomena parallel standard CL, the microscopic mathematical mechanisms driving them in ICL are fundamentally distinct. By explicitly decomposing the prediction error, we uncover counterintuitive dynamics unpredictable by classical CL.
>
> For example, Theorem 4.4 proves that catastrophic forgetting in ICL structurally arises from the attention mechanism's zero-sum task reweighting, not physical erasure, and is inherently order-dependent. Crucially, unlike standard paradigms where more historical data stabilizes performance, Theorem 4.3 reveals that simply extending context length can mathematically induce provable negative transfer, spiking generalization error when historical tasks are misaligned. Thus, our work provides the first mechanistic blueprint of how attention alone emulates and deviates from traditional CL.
>
> ---
>
> **Q3. Overly Restrictive Theoretical Framework: The theoretical analysis relies on a highly simplified model... The gap between the theoretical claims and the practical systems they aim to understand is simply too wide.**
>
> A3.We thank the reviewer for highlighting the gap between theoretical abstractions and nonlinear architectures. Our proofs focus on masked linear self-attention to achieve closed-form evaluations, a standard and necessary methodological step in recent rigorous ICL literature[1,2,3,4], allowing us to mathematically isolate the core aggregation behavior.
>
> Regarding the new technical difficulties: studying CL or ICL in isolation is insufficient. Traditional CL focuses on parameter-space interference via gradient updates, while single-task ICL assumes a homogeneous context. The unique challenge we resolve is proving how interference manifests entirely in the activation space without parameter updates. We mathematically demonstrate how a frozen, shared attention mechanism structurally induces forgetting through context competition and weight dilution, a mechanistic phenomenon completely invisible to classical CL and single-task ICL theories alike
>
> To bridge the gap to practical systems, our empirical validation extends far beyond synthetic GPT-2 experiments. As detailed in our Q1 response for Mgsi, we evaluated a modern 1.5B instruction-tuned model (Qwen2.5-1.5B-Instruct) on real-world NLP tasks (SST-2 and AG News). These empirical results perfectly mirror our linear model's predicted dynamics: severe negative transfer at short context lengths, recovery as relevant context expands, and massive forgetting. This confirms that the foundational mathematical baseline established by our simplified model successfully and robustly captures the dominant mechanisms governing realistic, state-of-the-art ICL systems.
>
> * [1] Ahn et al. "Transformers learn to implement preconditioned gradient descent for in-context learning." NeurIPS, 2023.
> * [2] Zhang et al. "Trained transformers learn linear models in-context." JMLR, 2024.
> * [3] Zhang et al. "In-context learning of a linear transformer block: Benefits of the MLP component and one-step GD initialization." NeurIPS, 2024
> * [4] Lu et al. "Asymptotic theory of in-context learning by linear attention." PNAS, 2025.

---

> > ### Author Rebuttal · Reviewer_pUNh · 2026-04-03
> >
> > The reviewer still has concerns about the expected insights. The paper reads more like a combination of continual learning and in-context learning (ICL). As noted in continual learning, catastrophic forgetting—such as the result in Theorem 4.4—is order-dependent for linear models. The authors are encouraged to develop beyond a simplified model, as this is largely a theoretical paper, while some  experiment was run to validate some effect regarding task order vs. forgetting. The reviewer keeps the same rating.

---

> > > ### Author Response · Authors · 2026-04-03
> > >
> > > We sincerely thank the reviewer for the continued engagement and for pushing us to clarify the significance of our theoretical insights. We understand concerns regarding the linear model's simplicity and parallels to classical continual learning (CL), but wish to clarify the necessity and significance of our approach.
> > >
> > > Establishing the first rigorous mathematical foundation for any new paradigm inherently requires starting with a tractable model. While forgetting is indeed a known phenomenon in classical CL, where it is driven by gradient updates, our work is the **first to theoretically prove that forgetting occurs entirely in the activation space without parameter updating**. This is a fundamental paradigm shift. By formalizing this, we provide the community with a mathematical blueprint showing that forgetting is an intrinsic, structural property of attention-based context aggregation, distinct from traditional parameter views. This theoretical baseline is crucial for understanding the limitations of long-context LLMs.
> > >
> > > To ensure these insights extend beyond simplified models, we explicitly designed two distinct levels of experimental validation: synthetic experiments on GPT-2 style models, and real-world NLP evaluations using a modern LLM (Qwen2.5-1.5B-Instruct). The fact that the massive forgetting, negative transfer, and order-dependency predicted by our simple linear theory perfectly manifest in this complex, state-of-the-art model proves that our derived insights successfully capture the dominant mechanisms of realistic, deep systems.
> > >
> > > While extending exact closed-form proofs to multi-layer, non-linear softmax Transformers remains a highly challenging open problem for the community. We view our current framework as the necessary foundational step and actively plan to explore extending these theoretical derivations to more complex architectures in our future work. We hope this clarifies why our approach was deliberately structured this way and how it provides a foundational, non-trivial contribution to the evolving understanding of LLMs.

---

### Official Review · Reviewer_GxmL · 2026-03-12

**Soundness:** 3
**Presentation:** 3
**Significance:** 3
**Originality:** 3
**Overall Recommendation:** 5
**Confidence:** 3

**Summary:**

This paper develops a theoretical framework for understanding In-context Continual Learning, where the LM needs to process a sequence of tasks, each with its own set of few-shot demonstrations, concatenated together in the same prompt. To make the theoretical analysis tractable, the authors use linear self-attention (masked) as a proxy for standard self-attention with causal masking. With the proxy LSA module, the authors provide a bias-variance-interference decomposition of the prediction error. They further test their predictions empirically with GPT-2 style models trained from scratch on synthetic linear regression tasks.

**Compliance With Llm Reviewing Policy:**

Affirmed.

**Final Justification:**

I am still a bit skeptical to what extent this paper's insights will actually transfer in real LMs, although the authors test them by training gpt-2 models.

However, I found this paper to be interesting, the theoretical insights very clean. I also think the authors' response to my questions/concerns were satisfactory.

**Key Questions For Authors:**

1. You perform your analysis on a linear self-attention module, which is a proxy for the standard self-attention with causal masking. Isn't the token-mixing in standard self-attention fundamentally different? The non-linear attention weights can create a more context dependent token-mixing. Also the whole analysis is based on a single LSA module. Standard transformer models have multiple layers of self-attention and MLPs. I am impressed that the insights hold at all for small GPT-2 style models. But I am curious if you think they will hold for larger real-world models.

2. The linear regression task you investigate, is it a very special case? In real in-context continual learning, I would assume the separation between the different tasks is more clear, and the tasks are more different from each other. Standard softmax self-attention will shine in this case, as it can attend to the relevant tokens and ignore the irrelevant ones. In the linear regression tasks, the attention module loses this advantage. Do you assume these insights would hold for more general language modeling tasks?

3. You train your own GPT-2 style models from scratch for the linear regression tasks to empirically test your theoretical insights. But you don't provide any additional details such as the number of layers, heads, the model dimension, etc. Also, what was the final performance of the trained models? Do your results hold for different model sizes (only checking 2-3 sizes will suffice)?

### Minor Issues:
* When you present Lemma 4.1, please explicitly refer to Appendix D where you provide the proof.

**Limitations:**

See my questions.

**Strengths And Weaknesses:**

**I am not a theory person and am probably not the most suitable reviewer for this paper. I request the AC to prioritize other reviewers ratings over mine while making the final decision.**

### Strengths:

* Clean theoretical analysis. I loved that the authors provided a bias-variance-interference decomposition of the error in Theorem 4.3.
* Theoretical insights built on a simpler linear self-attention module holds empirically for small GPT-2 style models with standard softmax attention. I found this very neat.
* I also think the theoretical insights intuitively makes sense.

### Weaknesses:
* I am a bit skeptical if these theoretical insights will hold for a larger real-world model on real language modeling tasks. The synthetic linear regression tasks seem a special case to me where the advantage of softmax attention is not fully utilized. Please see my questions for the authors for more details.

---

> ### Author Rebuttal · Authors · 2026-03-31
>
> **Q1. You perform your analysis on a linear self-attention...if you think they will hold for larger real-world models.**
>
> A1. We thank the reviewer for noting the architectural gap. We chose masked linear self-attention because it permits a closed-form error decomposition that isolates the fundamental dynamics of cross-task aggregation. While softmax attention creates context-dependent mixing, its core remains a normalized aggregation. As context expands with heterogeneous tasks, normalization inevitably forces a zero-sum competition for attention mass. Consequently, the non-linear network is structurally forced to dilute its focus, mimicking the exact linear interference and forgetting mechanisms we derived.
>
> To directly address whether these insights hold for real-world models, we conducted new evaluations using a modern instruction-tuned LLM (Qwen2.5-1.5B-Instruct, see Q1 from Mgsi). We evaluated multi-task behavior by testing the effect of context length. Empirical results exactly mirror our theory: evaluating Task B immediately after Task A reveals severe interference at short context lengths, gradually recovering through variance reduction as Task B's context expands. Evaluating the retention of Task A also shows substantial forgetting without parameter updates. This confirms our masked single-layer linear model captures foundational continual learning mechanisms that scale to massive LLMs.
>
> ---
> **Q2. The linear regression task.....Do you assume these insights would hold for more general language modeling tasks?**
>
> A2. We strongly agree with your intuition regarding the embedding space of SOTA models. Extensive pre-training likely maps distinct tasks to nearly orthogonal regions. Our framework anticipates this limit case: Theorem 4.4 explicitly proves that when historical tasks are statistically orthogonal to the target task, inter-task interference is minimized.
>
> However, we respectfully challenge the assumption that standard softmax self-attention completely bypasses interference simply because tasks are distinct. While orthogonality mitigates task similarity bias, the structural properties of attention-based aggregation persist. To empirically validate this for general language modeling, we systematically recorded predictions for Task A (sentiment analysis) and Task B (topic classification) within a single prompt. Despite these being highly distinct real-world NLP tasks, the Qwen2.5-1.5B softmax module did not flawlessly isolate them. As detailed in our Q1 response (Mgsi), we observed severe negative transfer at short context lengths, followed by intra-task recovery only as the relevant context expanded, alongside significant forgetting of Task A. Therefore, rather than being an isolated case, our linear analysis successfully captures the attention dilution and context competition governing general language tasks.
>
> ---
> **Q3. You train your own GPT-2...Do your results hold for different model sizes (only checking 2-3 sizes will suffice)?**
>
> A3. We thank the reviewer for pointing out this omission, and we are happy to provide the exact architectural details. In our experiments, we utilized a standard decoder-only GPT-2 architecture consisting of 12 layers, 8 attention heads, and an embedding dimension of 256 (~9.5M). We will update Appendix H in the revised manuscript to include a comprehensive table detailing all training hyperparameters, as well as the final converged training losses for these models.
>
> Regarding the scalability of our findings, we strongly agree that evaluating different model capacities strengthens our empirical claims. To address this, we conducted additional experiments across different parameter scales to verify our theory. Specifically, alongside the standard model used in our main text, we trained and evaluated a tiny GPT-2(3 layers, 2 heads, 64 embedding dimension, ~0.2M). Our results confirm that the observed generalization and forgetting patterns are consistent across these different model sizes. Due to space constraints in this rebuttal, we have provided an anonymous link [models_plot.pdf](https://anonymous.4open.science/r/iccl_re-3E66) illustrating the consistent impact of context length on generalization across these different architectures. The fully expanded results and detailed architectural tables will be fully integrated into the final version of the paper.

---

> > ### Author Rebuttal · Reviewer_GxmL · 2026-04-03
> >
> > I appreciate the authors for their response.
> > Congratulations on your good work. I felt like I learned something new from this paper. Looking forward to read the final version.

---

> > > ### Author Response · Authors · 2026-04-04
> > >
> > > We sincerely thank the reviewer for the positive feedback and for recognizing the value of our work. We are very glad that you found our paper insightful and that our rebuttal fully addressed your concerns. We deeply appreciate your time and constructive comments, which have helped us improve the final manuscript.

---

### Official Review · Reviewer_Mgsi · 2026-03-13

**Soundness:** 2
**Presentation:** 3
**Significance:** 2
**Originality:** 3
**Overall Recommendation:** 4
**Confidence:** 3

**Summary:**

The paper 's important contribution consists of introducing a theoretical framework for in-context continual learning, analyzing both generalization and forgetting in sequential task prompts under linear and masked linear self-attention. The paper also provides a bias–variance–interference decomposition, identifies how context length, task similarity, training prompt length, and task ordering affect performance, and supports their claims with experiments on GPT-style transformers trained on synthetic linear regression tasks.

**Compliance With Llm Reviewing Policy:**

Affirmed.

**Final Justification:**

thanks for the responses and they have solved my comments. I will modified the score to "weak accept". please see the strength details in the comment part.

**Key Questions For Authors:**

1. The paper is framed as a theory of in-context continual learning in LLMs, but both the theory and experiments focus on synthetic linear regression tasks. It's hard to fully support the intended scope of their claims: as a conceptual proof-of-principle for attention-based continual inference, or as a direct explanation of realistic multi-task prompting in LLMs. The authors need to verify their theory on more different task types to make it convincing.

2. Theorem 4.3 and Theorem 4.4 highlight context length, task similarity, and task ordering as key factors. It will be great if the author further analyze which factor is the most robust beyond the current linear-Gaussian setting, and which depends most strongly on the simplifying assumptions.

3. The paper argues that forgetting arises from reweighting of task contributions rather than from information loss. Can the authors elaborate on whether this interpretation is specific to masked linear attention, or whether they expect a similar decomposition to hold qualitatively for softmax-based Transformers?

4. In the experiments, task similarity is controlled through the angle between task means, and some tasks are described as benefiting from similarity to preceding tasks. The authors better provide a more systematic quantitative comparison between the theoretical interference terms and the observed MSE/forgetting curves, rather than mainly qualitative agreement.

5. The paper emphasizes that longer context does not always help and can even hurt when tasks are misaligned. The authors need to clarify whether this effect is mainly due to the specific averaging structure in the masked linear attention model, or whether they believe it reflects a broader principle of attention-based in-context adaptation.

**Limitations:**

I think the authors should explicitly state that the current results are derived in a simplified linear-regression setting with linear or masked linear self-attention, and that the empirical validation is restricted to GPT2-style models trained on synthetic Gaussian tasks.

**Strengths And Weaknesses:**

Strengths: The main strength is the theoretical framing. The paper formalizes in-context continual learning cleanly, defines both generalization and forgetting metrics, and derives explicit expressions showing how interference emerges from shared attention.

Weaknesses: The scope of the evidence is limited. Although the paper motivates the problem using LLMs and long heterogeneous prompts, both the theory and experiments are restricted to synthetic linear regression settings and simplified attention models. A second concern is that some claims in the experimental discussion are stronger than what the evidence seems to establish.

---

> ### Author Rebuttal · Authors · 2026-03-31
>
> **Q1. The paper... The authors need to verify their theory on more different task types to make it convincing.**
>
> A1.  We thank the reviewer for their valuable feedback. To further support our
> claims, we conduct additional experiments from two complementary perspectives.
>
> First, we validated our theory on complex function classes, including sparse linear regression and 2-layer ReLU networks $f(x)=\sum_{i=1}^r\alpha_i\sigma(w_i^\top x)$ in [Sparse_and_relu.png](https://anonymous.4open.science/r/iccl_re-3E66). Increasing context $M$ reduces Task 1 error (variance reduction), but for subsequent tasks, error eventually spikes (e.g., Sparse LR Task 5 hits 0.9905 at $M=19$), confirming that extended heterogeneous contexts inevitably induce systematic bias and negative transfer.
>
> Second, we evaluated a modern LLM (Qwen2.5-1.5B-Instruct) on sequential SST-2 (Task A) and AG News (Task B).
>
> | $M$ | Task B Acc | ICCL Task B | $\Delta$ | Parse Failures |
> |:-:|:-:|:-:|:-:|:-:|
> | 1 | 0.736 | 0.580 | - 0.156 | 123 |
> | 19 | 0.668 | 0.684 | + 0.016 | 61 |
>
> At small $M$, Task A biases attention, dropping Task B accuracy by 15.6% and severely disrupting instruction-following. As $M$ grows, intra-task information overcomes prior bias, recovering performance.
>
> | $M$ | Task A Acc | Final Task A (After Task B) | Forgetting $\Delta$ |
> |:-:|:-:|:-:|:-:|
> | 1 | 0.934 | 0.472 | **- 0.462** |
> | 19 | 0.922 | 0.536 | - 0.386 |
>
> Evaluating Task A retention after Task B reveals massive degradation (dropping from ~93% to ~47%). This confirms our theoretical finding: structural forgetting via task reweighting persists in modern Transformers, entirely without parameter updates.
>
> ---
>
> **Q2. analyzes which factor is the most robust beyond the current linear-Gaussian setting, and which depends most strongly on the simplifying assumptions.**
>
> A2. We thank the reviewer for this insightful question. Context length and task ordering are highly robust beyond linear-Gaussian settings, as they stem from the shared attention mechanism itself. Expanding context inevitably aggregates historical information, causing interference (validated by our Qwen2.5 experiments). Conversely, task similarity depends more on our synthetic construction; in real LLMs, massive pretraining maps heterogeneous tasks to nearly orthogonal representations, making the smooth transfer-to-interference transition less pronounced in practice.
>
> ---
>
> **Q3. Can the authors elaborate ... for softmax-based Transformers?**
>
> A3. This decomposition holds qualitatively for softmax Transformers. Unlike standard CL, where parameter updates cause information loss, ICCL perfectly preserves sequences in the context window. Forgetting must intrinsically arise from attention aggregation. Softmax normalization dictates a zero-sum competition for probability mass. Appending new tasks expands the softmax denominator, diluting weights assigned to earlier tasks. This non-linear reweighting functionally mirrors our derived linear reweighting, causing identical structural interference, as validated by our GPT-2 experiments.
>
> ---
>
> **Q4. The authors better provide a more systematic quantitative comparison between the theoretical interference terms and the observed MSE/forgetting curves, rather than mainly qualitative agreement.**
>
> A4. We agree that a systematic quantitative comparison could strengthen our claims.
> [T_and_E.png](https://anonymous.4open.science/r/iccl_re-3E66)
> While theoretical calculations predict steady generalization error decreases (L260-274), our GPT-2 empirical results explicitly capture the competing bias-variance trade-off parametrized in Theorem 4.3. For subsequent tasks (Tasks 2-5), empirical error exhibits significant spikes at intermediate context lengths (e.g., Task 5 peaks at $M=3$) before recovering. This structural inflection quantifies Section 4.1: expanding misaligned contexts initially injects systematic bias that overwhelms variance reduction until intra-task information dominates.
>
> ---
>
> **Q5. The paper emphasizes that longer context does not always help and can even hurt when tasks are misaligned. The authors need to clarify whether this effect is mainly due to the specific averaging structure in the masked linear attention model, or whether they believe it reflects a broader principle of attention-based in-context adaptation.**
>
> A5. We believe that this degradation reflects a fundamental principle of attention-based adaptation, not merely a linear attention model. In softmax attention, normalization dictates a zero-sum competition. Extending prompts with misaligned tasks forces the mechanism to assign non-zero weights to irrelevant historical tokens, diluting attention mass for the target task and synthesizing conflicting representations. Our GPT-2 and Qwen2.5-1.5B empirical results both confirm that extending misaligned prior contexts consistently degrades target accuracy, demonstrating this structural interference generalizes fully to multi-layer softmax Transformers.

---

> > ### Author Rebuttal · Reviewer_Mgsi · 2026-04-03
> >
> > thanks for the responses and they have solved my comments.

---

> > > ### Author Response · Authors · 2026-04-04
> > >
> > > We sincerely thank the reviewer for their time, constructive feedback, and for carefully considering our rebuttal. We are glad that our additional explanations and experiments have fully resolved your concerns. We deeply appreciate your time and your contribution to improving our paper.

---

### Official Review · Reviewer_SvmR · 2026-03-16

**Soundness:** 3
**Presentation:** 3
**Significance:** 3
**Originality:** 3
**Overall Recommendation:** 4
**Confidence:** 2

**Summary:**

The paper studies the theoretical aspect of in-context continual learning with masked linear self-attension. The paper derives prediction error in Theorem 4.3 and provides mathematical intuition of the three components of the error. Analyzing the formulation of the error, the paper summarizes the dynamixs/takeaways in Table 1. Finally, the paper provides numerical experiments supporting theortical findings.

**Compliance With Llm Reviewing Policy:**

Affirmed.

**Key Questions For Authors:**

I did not see how training phase (the training of self-attention) affects the prediction error in Theorem 4.3. How do we understand the effect of training on the error?

**Strengths And Weaknesses:**

Strength:

1. Most parts of the paper is very clear. The paper carefully introduces the masked linear-self attention and how is the prompt of in-context continual learning is composed.
2. The paper derives the prediction error for task t, coupled with explanations of terms and analyses on variables, giving intuition behind the terms.
3. Analyzing in-context continual learning is new to me.

Weakness:

1. Connecting formulation (1) and formulation line183, does **P** contains multiple columns with x only?
2. The main paper does not introduce the training phase and number of layers of attention.
3. In experiments, figures are too tiny to observe.

---

> ### Author Rebuttal · Authors · 2026-03-31
>
> **Q1. I did not see how the training phase (the training of self-attention) affects the prediction error in Theorem 4.3. How do we understand the effect of training on the error?**
>
> A1. In our theoretical analysis, the effect of training is primarily reflected in the final learned parameter matrix, which we characterize in Theorem A.2. As discussed in Section 4.1, the dominant factor influencing this process is the training samples. Taking generalization as an example, in Section 4.1 we provide a detailed analysis of how the number of samples affects model performance. In general, larger sample sizes do not always lead to better performance. In our experimental observations, however, with the number of training steps set to 500k, the final loss tends to stabilize regardless of the number of training samples (as long as it exceeds 10).  To further support our theoretical findings, the reviewer may refer to [overall.pdf](https://anonymous.4open.science/r/iccl_re-3E66), which shows that in curriculum learning, the loss curve initially decreases and then increases as the number of samples grows.
>
> We also emphasize that the obtained parameter is derived under gradient flow dynamics with appropriate initialization, yielding a closed-form analytical result. In practical training scenarios, however, the learning process is affected by additional factors, such as the learning rate, model size, and the number of attention heads, etc.
>
> ---
>
> **Q2. Connecting formulation (1) and formulation line 183, does P contain multiple columns with x only? The main paper does not introduce the training phase and number of layers of attention. In experiments, figures are too tiny to observe.**
>
> A2. We thank the reviewer for the insightful question. We need to clarify that each task in the P consists of two types of samples: (1) input–label pairs, which serve as in-context examples, and (2) input-only samples, which act as queries. In our theoretical analysis, we adopt a single-attention setting for tractability. In contrast, to better align with realistic scenarios, our experiments use GPT-2 Standard as the backbone model, which contains 12 Transformer layers, 8 attention heads, and an embedding dimension of 256.
>
> We appreciate the reviewer’s note on figure readability. Due to space constraints, some figures in the main paper are necessarily compact. We will enlarge the relevant figures and improve their readability in the revised version.

---

> > ### Author Rebuttal · Reviewer_SvmR · 2026-04-07
> >
> > My questions are partially solved. I do not have a good understanding of the paper as noted by "Confidence: 2", so I keep my current score.

---

> > > ### Author Response · Authors · 2026-04-08
> > >
> > > Thank you for reviewing our rebuttal and your kind acknowledgement. Since you mentioned having follow-up questions, please feel free to share them with us. We are more than happy to address any remaining concerns and provide further clarifications to help with your understanding of our paper. Thank you again for your time and effort.

---

### Decision · Program_Chairs · 2026-04-30

**Decision:**

Accept (regular)

**Comment:**

This paper develops a theoretical framework for in-context continual learning, analyzing how a pretrained Transformer handling multiple sequential tasks within a single prompt exhibits both generalization and forgetting through shared attention, and derives a bias–variance–interference decomposition under linear and masked linear self-attention. The reviewers generally viewed the paper positively, highlighting the novelty of formalizing continual learning entirely at inference time without parameter updates, the clarity of the theoretical development, and the supporting empirical validation; the overall recommendations are 5/4/4/4. The main concerns were that the theory relies on simplified linear attention and synthetic linear-regression settings, that some claims initially seemed broader than the evidence, and that the connection to realistic large language models needed to be strengthened. In the rebuttal, the authors responded constructively by clarifying that their contribution is the first theoretical account of forgetting arising from attention-based aggregation rather than weight overwriting, by adding further experiments on more complex functions and on a modern instruction-tuned LLM (Qwen2.5-1.5B-Instruct) on real NLP tasks, and by providing more architectural and training details; these responses were viewed favorably, with reviewers indicating that most or all concerns were resolved and some explicitly raising their scores. Overall, I find that the paper makes a meaningful and timely theoretical contribution at the intersection of in-context learning and continual learning, and I recommend Accept.